# Disrupted PGR-B and ESR1 signaling underlies defective decidualization linked to severe preeclampsia

**Tamara Garrido-Gomez**[1]*[†], **Nerea Castillo-Marco**[1][†], **Mónica Clemente-Ciscar**[2][†], **Teresa Cordero**[1], **Irene Muñoz-Blat**[1], **Alicia Amadoz**[2], **Jorge Jimenez-Almazan**[2], **Rogelio Monfort-Ortiz**[3], **Reyes Climent**[3], **Alfredo Perales-Marin**[3,4], **Carlos Simon**[1,4,5]*

[1]Igenomix Foundation, INCLIVA, Valencia, Spain; [2]Igenomix, Valencia, Spain; [3]Department of Obstetrics and Gynecology, University and Polytechnic La Fe Hospital, Valencia, Spain; [4]Department of Obstetrics and Gynecology, School of Medicine, Valencia University, Valencia, Spain; [5]Obstetrics & Gynecology, BIDMC Harvard University, Boston, United States

**\*For correspondence:**
tamara.garrido@
igenomixfoundation.com (TG-G);
carlos.simon@uv.es (CS)

[†]These authors contributed equally to this work

## Abstract

**Background:** Decidualization of the uterine mucosa drives the maternal adaptation to invasion by the placenta. Appropriate depth of placental invasion is needed to support a healthy pregnancy; shallow invasion is associated with the development of severe preeclampsia (sPE). Maternal contribution to sPE through failed decidualization is an important determinant of placental phenotype. However, the molecular mechanism underlying the in vivo defect linking decidualization to sPE is unknown.

**Methods:** Global RNA sequencing was applied to obtain the transcriptomic profile of endometrial biopsies collected from nonpregnant women who suffer sPE in a previous pregnancy and women who did not develop this condition. Samples were randomized in two cohorts, the training and the test set, to identify the fingerprinting encoding defective decidualization in sPE and its subsequent validation. Gene Ontology enrichment and an interaction network were performed to deepen in pathways impaired by genetic dysregulation in sPE. Finally, the main modulators of decidualization, estrogen receptor 1 (*ESR1*) and progesterone receptor B (*PGR-B*), were assessed at the level of gene expression and protein abundance.

**Results:** Here, we discover the footprint encoding this decidualization defect comprising 120 genes—using global gene expression profiling in decidua from women who developed sPE in a previous pregnancy. This signature allowed us to effectively segregate samples into sPE and control groups. *ESR1* and *PGR* were highly interconnected with the dynamic network of the defective decidualization fingerprint. *ESR1* and *PGR-B* gene expression and protein abundance were remarkably disrupted in sPE.

**Conclusions:** Thus, the transcriptomic signature of impaired decidualization implicates dysregulated hormonal signaling in the decidual endometria in women who developed sPE. These findings reveal a potential footprint that could be leveraged for a preconception or early prenatal screening of sPE risk, thus improving prevention and early treatments.

**Funding:** This work has been supported by the grant PI19/01659 (MCIU/AEI/FEDER, UE) from the Spanish Carlos III Institute awarded to TGG. NCM was supported by the PhD program FDGENT/2019/008 from the Spanish Generalitat Valenciana. IMB was supported by the PhD program PRE2019-090770 and funding was provided by the grant RTI2018-094946-B-100 (MCIU/AEI/FEDER, UE) from the Spanish Ministry of Science and Innovation with CS as principal investigator. This research was funded partially by Igenomix S.L.

## Introduction

Preeclampsia (PE) is a severe complication of late pregnancy and is the second leading cause of maternal mortality in the US, affecting 8% of first-time pregnancies (*Gifford, 2000*). PE is characterized by the onset of hypertension, proteinuria, and other signs of maternal vascular damage that contributes to maternal and neonatal mortality and morbidity (*Gifford, 2000*). Severe preeclampsia (sPE) is diagnosed based on elevated blood pressure (systolic ≥160 or diastolic of ≥100 mm Hg) or thrombocytopenia, impaired liver function, progressive renal insufficiency, pulmonary edema, or the onset of cerebral or visual disturbances (*Gynecologists ACoOa, Pregnancy TFoHi, 2013*). sPE is a placental insufficiency syndrome mediated by early-deficient extravillous trophoblast (EVT) invasion of uterine decidua and spiral arterioles, leading to incomplete endovascular invasion and altered uteroplacental perfusion (*Roberts and Cooper, 2001*; *Brosens et al., 2019*; *Staff et al., 2020*). Why shallow EVTs invasion occurs, however, remains to be determined (*Fisher, 2015*).

Pregnancy health is dictated by the embryo, placenta, and the quality of the maternal decidua, where EVTs invasion and remodeling of maternal spiral arteries occur (*Norwitz et al., 2001*; *Cha et al., 2012*). Accumulated evidence suggests that the contribution of the decidua to the etiology of PE (*Rabaglino et al., 2015*), sPE (*Garrido-Gomez et al., 2017*; *Garrido-Gomez et al., 2020*; *Garrido-Gómez et al., 2020*), and placenta accreta (*McNally et al., 2020*) is significant, and cellular signaling in the decidua may determine whether these conditions develop. Decidualization is the remodeling of the endometrium initiated after ovulation that is necessary for adequate trophoblast invasion and subsequent placentation (*Gellersen and Brosens, 2014*). Defective decidualization (DD) entails the inability of the endometrial compartment to undertake tissue differentiation, leading to aberrations in placentation and compromising pregnancy health (*Garrido-Gómez et al., 2020*).

In humans and other great apes, the formation of the decidua is a conceptus-independent process driven by progesterone and the second messenger cyclic adenosine monophosphate (*Brar et al., 1997*) that stimulates synthesis of a complex network of intracellular and secreted proteins through progesterone receptor (PR) activation. Endometrial decidualization involves secretory transformation of uterine glands (*Kelleher et al., 2018*), influx of specialized immune cells, vascular remodeling, and morphological (*Dunn et al., 2003*; *Ramathal et al., 2010*), biochemical (*Giudice et al., 1998*; *Jabbour and Critchley, 2001*), and transcriptional reprogramming of the endometrial stromal compartment (*Wang et al., 2020*). We recently characterized the transcriptomics of human decidualization at single-cell resolution from secretory endometrial samples and showed that the process is initiated gradually after ovulation, with a direct interplay between stromal fibroblasts and lymphocytes (*Wang et al., 2020*). However, most knowledge on decidual function in health and disease comes from in vitro model systems (*Ng et al., 2020*; *Zhou et al., 2013*; *Ganeff et al., 2009*).

In the present study, we aimed to discern the preconception decidual transcriptomic signature associated with in vivo DD. We performed a comparative global transcriptional profiling of endometrium in women who developed sPE in a previous pregnancy. Initially, we identified 593 genes differentially expressed in sPE compared to control cases. Then, molecular DD fingerprint of 120 genes associated with the development of sPE was defined and evaluated as a diagnostic tool in an independent cohort of samples. Finally, we identified PR and estrogen receptor 1 (ER1) as targets of the DD fingerprint genes. Our findings indicate that an endometrial transcriptomic signature persists years after the affected pregnancy. This signature may be leveraged for a preconception or early prenatal screening strategy in assessing sPE risk and may inform the development of sPE therapies.

## Methods

**Key resources table**

| Reagent type (species) or resource | Designation | Source or reference | Identifiers | Additional information |
|---|---|---|---|---|
| Biological sample (*Homo sapiens*) | Endometrial biopsies | University and Polytechnic La Fe Hospital (Valencia, Spain) | | Freshly isolated from human donors |
| Antibody | Anti-progesterone receptor antibody [YR85] (rabbit monoclonal anti-human) | Abcam | Cat: AB32085 RRID:AB_777452 | Dilution: (1:50) |

*Continued on next page*

*Continued*

| Reagent type (species) or resource | Designation | Source or reference | Identifiers | Additional information |
|---|---|---|---|---|
| Antibody | Anti-estrogen receptor alpha antibody (mouse monoclonal antibody) | Santa Cruz | Cat: sc-8002 RRID:AB_627558 | Dilution: (1:50) |
| Antibody | Goat anti-rabbit IgG H&L (Alexa Fluor 488) (goat polyclonal) | Abcam | Cat: ab150077 RRID:AB_2630356 | Dilution: (1:1000) |
| Antibody | Goat anti-mouse IgG (H + L) Cro Alexa Fluor 488 (goat polyclonal) | Invitrogen | Cat: A-11001 RRID:AB_2534069 | Dilution: (1:1000) |
| Sequence-based reagent | RT-qPCR primers | This paper | | *Supplementary file 3* |
| Commercial assay or kit | QIAsymphony RNA Kit | Qiagen | 931636 | Global RNA-seq library preparation |
| Commercial assay or kit | Illumina TruSeq Stranded mRNA sample prep kit | Illumina | 20020595 | Global RNA-seq library preparation |
| Commercial assay or kit | Kapa SYBR fast qPCR kit | Kapa Biosystems Inc | KK4602 | Global RNA-seq library preparation |
| Commercial assay or kit | TruSeq RNA CD Index Plate (96 indexes, 96 samples) | Illumina | 20019792 | RNA sequencing |
| Commercial assay or kit | NextSeq 500/550 cartridge of 150 cycles | Illumina | FC-404-2002 | RNA sequencing |
| Commercial assay or kit | SuperScript VILO cDNA Synthesis Kit | Thermo Fisher Scientific | 11754250 | RT-qPCR. cDNA preparation |
| Software, algorithm | STAR | *Dobin et al., 2013* | URL: http://code.google.com/p/rna-star/ RRID:SCR_004463 | RNA-seq analysis Read aligner Version 2.4.2a |
| Software, algorithm | FastQC | | URL: http://www.bioinformatics.babraham.ac.uk/projects/fastqc/ RRID:SCR_014583 | RNA-seq analysis Quality of FASTQ file determination Version 0.11.2 |
| Software, algorithm | SAMtools | *Li et al., 2009* | URL: http://htslib.org/ RRID:SCR_002105 | RNA-seq analysis SAM and BAM manipulation files Version 1.1 |
| Software, algorithm | HTSeq | *Anders et al., 2015* | URL: http://htseq.readthedocs.io/en/release_0.9.1/ RRID:SCR_005514 | RNA-seq analysis To count the number of reads per gene Version 0.6.1p1 |
| Software, algorithm | BEDtools | *Quinlan and Hall, 2010* | URL: https://github.com/arq5x/bedtools2 RRID:SCR_006646 | RNA-seq analysis To obtain gene coverage Version 2.17.0 |
| Software, algorithm | edgeR | *Robinson et al., 2010* | URL: http://bioconductor.org/packages/edgeR/ RRID:SCR_012802 | RNA-seq analysis To analyze differentially expressed genes Version 3.24.3 |
| Software, algorithm | String | *Jensen et al., 2009* | URL: http://string.embl.de/ RRID:SCR_005223 | Interaction Network. |
| Software, algorithm | Cytoscape | *Shannon et al., 2003* | URL: http://cytoscape.org SCR_003032 | Interaction Network |
| Software, algorithm | CytoHubba | *Chin et al., 2014* | URL: http://apps.cytoscape.org/apps/cytohubba RRID:SCR_017677 | Interaction Network |
| Other | Custom scripts | | URL: https://github.com/mclemente-igenomix/garrido_et_al_2021 | The specific script to run RNA-seq analysis |

## Study design

A total of 40 non-pregnant women who experienced a previous pregnancy were enrolled in this study for endometrial RNA-sequencing analysis. Endometrial samples were obtained for research purposes during late secretory phase in 24 women who had developed sPE in a previous pregnancy and in 16 women with no history of sPE with full term (n = 8) and preterm pregnancies (n = 8) as controls. sPE was clinically defined based on elevated blood pressure (systolic ≥160 or diastolic of ≥100 mm Hg) or thrombocytopenia, impaired liver function, progressive renal insufficiency, pulmonary edema, or the onset of cerebral or visual disturbances. Endometrial biopsies were processed to obtain RNA and then converted to cDNA for library generation to perform next-generation sequencing. The experimental design was based on a stratified random sampling with a 70:30 proportion in two cohorts: a training (n = 29) and validation (n = 11) set of samples. The training set of samples was analyzed by RNA-seq to identify the global transcriptomic profiling changes between control (n = 12) and sPE (n = 17) samples. Selection criteria were applied to define a transcriptomic fingerprinting associated with DD detected in sPE. Finally, targeted analysis of the DD signature was validated in the test set composed of controls (n = 4) and sPE (n = 7).

## Human donors

Endometrial samples were collected from women aged 18–42 without any medical condition who had been pregnant 1–8 years earlier. All participants had regular menstrual cycles (26–32 days) with no underlying gynecological pathological conditions and had not received hormonal therapy in the 3 months preceding sample collection. After the inclusion criteria were applied, endometrial biopsies were obtained by pipelle catheter (Genetics Hamont-Achel, Belgium) under sterile conditions in the late secretory phase (cycle days 22–32). Specimens were kept in preservation solution until processing. Maternal and neonatal characteristics of women with sPE and controls are summarized in *Supplementary file 1*. Biological and technical variables for each donor were considered to discard confounding effects on the transcriptomic profile (*Supplementary file 2*). This study was approved by the Clinical Research Ethics Committee of University and Polytechnic La Fe Hospital (Valencia, Spain; 2011/0383), and written informed consent was obtained from all participants before tissue collection and all samples were anonymized.

## RNA extraction

Total RNA from endometrial biopsies was isolated using QIAsymphony RNA kit (Qiagen, Hilden, Germany) following the manufacturer's protocol. RNA concentrations were quantified using a Multiskan GO spectrophotometer (Thermo Fisher Scientific, Waltham, USA) at a wavelength of 260 nm. Integrity of the total RNA samples was evaluated by the RNA integrity number (RIN) and DV200 metrics using an Agilent high-sensitivity RNA ScreenTape in a 4200 TapeStation system (Agilent Technologies Inc, Santa Clara, CA). Samples used for the global RNA-seq showed RIN values ranging from 4.9 to 9.2.

## Global RNA-seq library preparation and transcriptome sequencing

cDNA libraries from total RNA samples (n = 40) were prepared using an Illumina TruSeq Stranded mRNA sample prep kit (Illumina, San Diego, CA) following a balanced batch-group design. 3 µg of total RNA were used as the RNA input according to the manufacturer's protocol. mRNAs were isolated from the total RNAs by purifying the poly-A containing molecules using poly-T oligo attached to magnetic beads. The RNA fragmentation, first- and second-strand cDNA syntheses, end repair, single 'A' base addition, adaptor ligation, and PCR amplification were performed according to the manufacturer's protocol. The average size of the cDNA libraries was approximately 350 bp (including the adapters). cDNA libraries were quantified using an Agilent D1000 ScreenTape in a 4200 TapeStation system (Agilent Technologies Inc). Libraries were normalized to 10 nM and pooled in equal volumes. The pool concentration was quantified by qPCR using the KAPA Library Quantification Kit (Kapa Biosystems Inc) before sequencing in a NextSeq 500/550 cartridge of 150 cycles (Illumina). Indexed and pooled samples were sequenced 150 bp paired-end reads by on the Illumina NextSeq 500/550 platform according to the Illumina protocol.

## RNA-seq analysis

Reads were mapped to the hg19 human genome transcriptome using the STAR (version 2.4.2 a) read aligner (*Dobin et al., 2013*). FastQC (version 0.11.2) was used to determine the quality of FASTQ files. The manipulation of SAM and BAM files was done with the software SAMtools (version 1.1) (*Li et al., 2009*). To count the number of reads that could be assigned to each gene, we used HTSeq (version 0.6.1p1; *Anders et al., 2015*) and BEDtools software (version 2.17.0; *Quinlan and Hall, 2010*) to obtain gene coverage and work with bedFiles. Quality control filters in each program were used following the software package recommendations, and reads were filtered by mapping quality greater than 90% . Transcriptomic data were deposited in the Gene Expression Omnibus database (accession number GSE172381). The Bioconductor package edgeR (version 3.24.3; *Robinson et al., 2010*) was used to analyze differentially expressed genes (DEGs). The trimmed mean of M-values normalization method was applied to our gene expression values. The glmTreat function was used to find DEGs between groups. The p-value adjustment method was false discovery rate (FDR) with a cutoff of 0.05 (FDR < 0.05) and the fold-change (FC) threshold was 1.2. edgeR analysis was carried out in R version 3.5.1. A volcano plot was created to visualize DEGs. Custom scripts are available on GitHub at link https://github.com/mclemente-igenomix/garrido_et_al_2021.

## Transcriptomic fingerprinting definition and validation

Genes with assigned EntrezID with an FDR cutoff of 0.05 and an expression ≥1.4- fold higher in the sPE vs. control training set samples were selected to define a fingerprint associated with DD in sPE. Targeted analysis of fingerprinting genes was performed using the validation set of samples. PCA and unsupervised hierarchical clustering with a Canberra distance based on gene signature were performed comparing sPE to control specimens. Custom scripts are available on GitHub at https://github.com/mclemente-igenomix/garrido_et_al_2021.

## Enrichment analysis

Gene Ontology (GO) analyses were conducted to obtain biological processes using the *goana* function in edgeR (*Robinson et al., 2010*). The input genes were those 120 included in the fingerprinting (*Figure 3—source data 1*). The p-value adjustment method was FDR with a cutoff of 0.05 (FDR < 0.05; *Figure 3—source data 2*). Custom scripts are available on GitHub at https://github.com/mclemente-igenomix/garrido_et_al_2021.

## Interaction network

An interaction network between proteins encoded by DD fingerprinting genes was created using the functional analysis suite String (*Jensen et al., 2009*). To construct the network, the interactions included were from curated databases and included experimentally determined and predicted interactions, textmining, co-expression information. The clustering algorithm k-means was applied based on the distance matrix obtained from the String global scores. The network was visualized using Cytoscape software (*Shannon et al., 2003*). Hub genes were extracted using the maximal clique centrality (MCC) and maximum neighborhood component (MNC) of the cytoHubba plugin (*Chin et al., 2014*). The overlapping genes identified by the two topological analysis methods were selected as the hub genes.

## qRT-PCR

Gene expression of *IHH*, *MSX2*, *ESR1,* and *PGR* isoforms in the endometrial tissue from a subset of women with prior sPE (n = 13) compared to controls (n = 9) was obtained by RT-qPCR. Specific primers for each gene are described in *Supplementary file 3*. cDNA was generated from 400 ng of RNA using the SuperScript VILO cDNA Synthesis Kit (Thermo Fisher Scientific). Template cDNA was diluted 5 in 20 and 1 µL was used in each PCR. Real-time PCR was performed in duplicate in 10 µL using commercially validated Kapa SYBR fast qPCR kit (Kapa Biosystems Inc, Basilea, Switzerland) and the Lightcycler 480 (Roche Molecular Systems, Inc, Pleasanton, CA) detection system. Samples were run in duplicate along with appropriate controls (i.e., no template, no RT). Cycling conditions were as follows: 95 °C for 3 min, 40 cycles of 95 °C for 10 s, 60 °C for 20 s, and 72 °C for 1 s. A melting curve was done following the product specifications. Data were analyzed using the comparative Ct method (2−ΔΔCT). Data were normalized to the housekeeping gene β-actin, and changes in gene expression

were calculated using the ΔΔCT method with the control group used as the calibrator; values are illustrated relative to median in the control group. The relative expression of *PGR-A* mRNA was calculated by subtracting the relative expression of *PGR-B* mRNA from that of *PGR* total.

## Immunofluorescence of tissue sections

Endometrial tissue samples were fixed in 4% paraformaldehyde and preserved in paraffin-embedded blocks. For immunostaining, tissue sections were deparaffinated and rehydrated. Antigen retrieval was performed with buffer citrate 1× at 100 °C for 10 min. Then, non-specific reactivity was blocked by incubation in 5% BSA/0.1% PBS-Tween 20 at room temperature for 30 min. Sections were incubated at room temperature for 1.5 hr with primary antibodies (1:50 rabbit monoclonal anti-human progesterone receptor, Abcam, Cambridge, UK) and 1:50 mouse monoclonal anti-human estrogen receptor 1 (Santa Cruz Biotechnology, CA) diluted in 3% BSA/0.1% PBS-Tween 20. Then, slides were washed two times for 10 min with 0.1% PBS-Tween 20 before they were incubated for 1 hr at room temperature with AlexaFluor-conjugated secondary antibodies diluted in 3% BSA/0.1% PBS-Tween 20 (1:1000). Finally, slides were washed two times in 0.1% PBS-Tween 20. To visualize nuclei, 4',6-diamidino-2-phenylindole at 400 ng/μL was used. Tissue sections were examined using a EVOS M5000 microscope.

## Statistical analysis

Clinical data are expressed as mean ± standard error mean (SEM). Clinical data were evaluated by Wilcoxon test for comparisons between sPE and control samples. Statistical significance was set at p<0.05. Differential expression analysis was performed using the R package edgeR.

## Results
### Endometrial transcriptome alterations during decidualization in sPE

To identify transcriptomic alterations during decidualization in sPE, we applied global RNA sequencing (RNA-seq) to endometrial biopsies obtained in the late secretory phase from women who developed sPE in a previous pregnancy (n = 24) and controls who never had sPE (n = 16) (GSE172381). Clinical maternal and neonatal characteristics of the participants are summarized in *Supplementary file 1.* After quality trimming and filtering, reads were aligned to the reference genome hg19. The 40 samples produced 56,638 raw sequencing genes; after normalization, 18,301 genes were included in the analysis. Biological and technical variables for each donor were considered to discard confounding effects on the transcriptomic profile (*Supplementary file 2*). Controls included women who had a preterm birth with no signs of infection (n = 8) and women who gave birth at full term with normal obstetric outcomes (n = 8). Transcriptomic profiles were compared by differential expression analysis, revealing no significant changes in the endometrial transcriptome between preterm and term controls (FDR ≥ 0.05; *Figure 1—figure supplement 1A*). Principal component analysis (PCA) supported that there was no underlying pattern of distribution depending on gestational age at delivery (*Figure 1—figure supplement 1B*). Once we ruled out bias on controls, we randomly split samples into two cohorts, a training set (70%) and a test set (30%) (*Figure 1A*). Random sampling occurred within each class (sPE and controls), so overall class distribution of the data was preserved. The training set (n = 29) was used for the identification of molecular fingerprinting encoding DD in sPE, while the test set (n = 11) was used to confirm our findings. All samples in both cohorts were processed and sequenced in the same manner.

Transcriptional analysis in the training set was performed by comparing gene expression patterns in sPE (n = 17) and controls (n = 12). This comparison revealed 593 DEGs based on FDR < 0.05 and with at least 1.2 FC between groups (FC ≥ 1.2). DEGs are shown in the volcano plot through yellow dots (*Figure 1B*). A total of 155 upregulated and 438 downregulated DEGs were identified as being associated with DD in sPE (*Figure 1C*; complete list in *Figure 1—source data 1*). Downregulated transcripts include those involved in decidualization, such as *MMP3*, *PRL*, *IL-6*, and *IHH*; and genes associated with signaling (e.g., *NR4A3* and *IL8*), growth factors (e.g., *FGF1* and *FGF7*), angiogenesis (e.g., *EDN2* and *TMEM215*), and immune response (*CCL20*, *CXCL3*, and *IGHG1*). Upregulated genes are involved in amino acid metabolic/catabolic processes (*IDO2* and *CAPN3*), transport, and oxidoreductase activity.

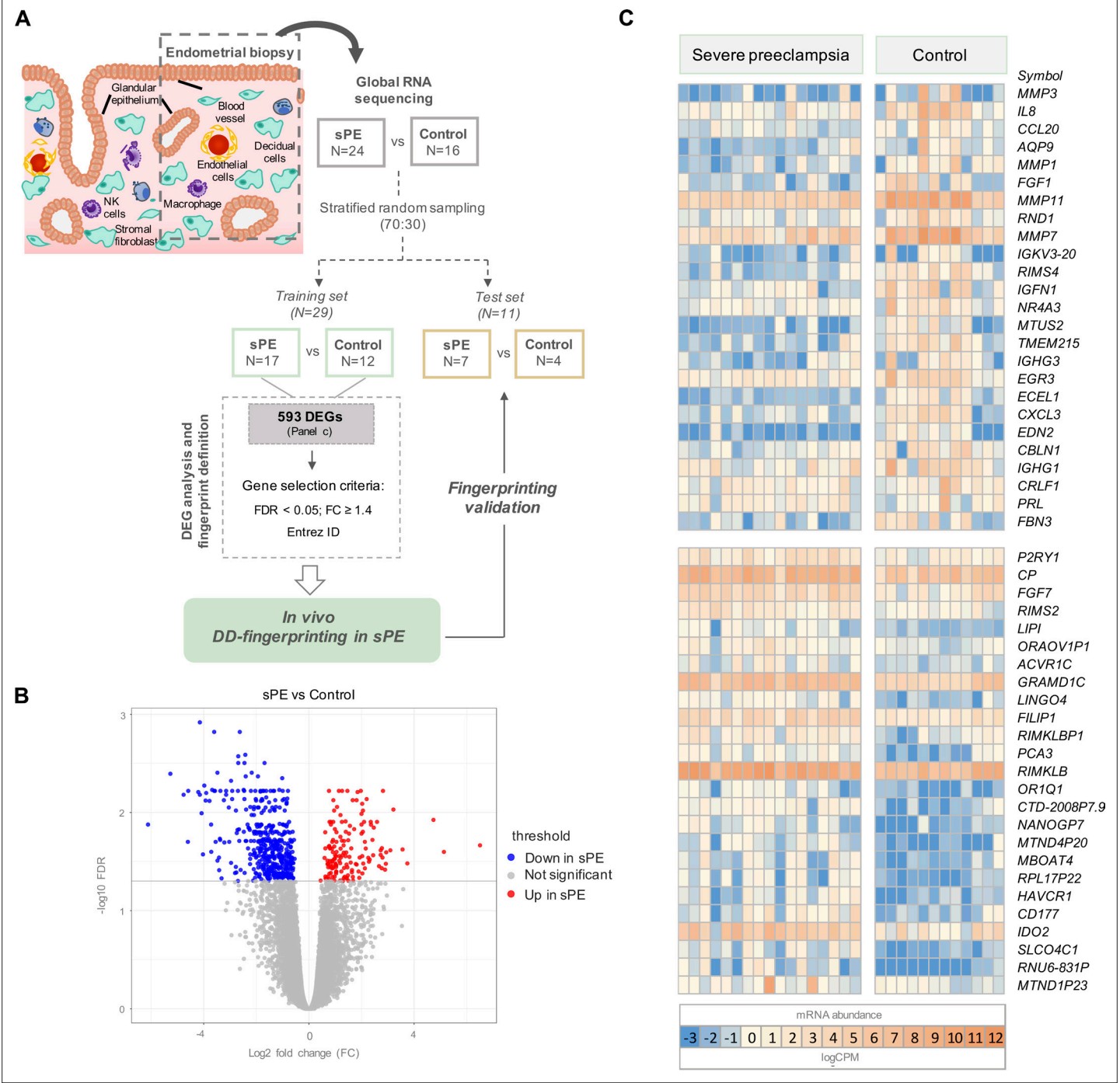

**Figure 1.** Global RNA-seq transcriptomic results revealed 593 differentially expressed genes (DEGs) in severe preeclampsia (sPE) vs. control samples. (**A**) Schematic drawing of the study design used to identify and validate defective decidualization (DD) fingerprinting in sPE. (**B**) Statistical significance (-log10 FDR) vs. gene expression log2 fold change (FC) is displayed as a volcano plot of global RNA-seq results. Label indicates: downregulated in sPE (*blue dots*); upregulated in sPE (*red dots*); not significant genes (*grey dots*). (**C**) Heatmap showing the 25 most upregulated and downregulated genes (total = 593; *Figure 1—source data 1*) of control vs. sPE samples. See also *Figure 1—source data 1*.

The online version of this article includes the following figure supplement(s) for figure 1:

**Source data 1.** The 593 statistically differentially expressed genes (false discovery rate [FDR] < 0.05) with at least 1.2-fold change (FC ≥ 1.2) in severe preeclampsia (sPE) vs. control cases obtained from RNA-seq analysis.

**Figure supplement 1.** Transcriptomic analysis based on gestational age at delivery of control samples.

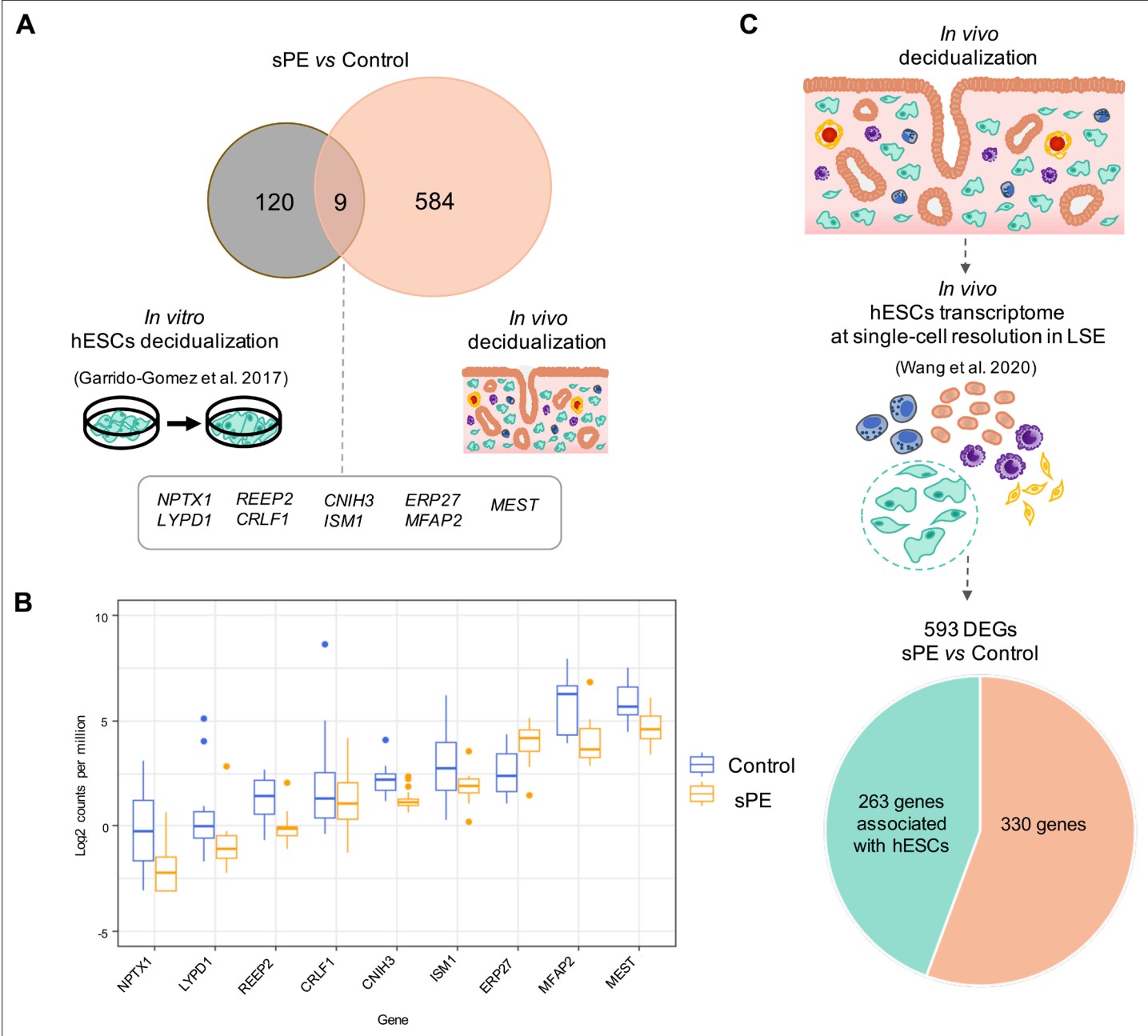

**Figure 2.** Defective decidualization (DD) transcriptomics in vitro vs. in vivo. (**A**) Common genes between previous in vitro (left) and current in vivo approaches analyzing decidualization (right). Nine genes overlap in both approaches. (**B**) Box plot showing the average expression of the nine common genes between control (blue boxes) and severe preeclampsia (sPE) (orange boxes) samples. (**C**) From the 593 differentially expressed genes (DEGs) obtained by global RNA-seq, a subset of 263 DEGs were identified as genes with a human endometrial stromal cell (hESC) origin using the scRNA-seq data published by *Wang et al., 2020*. See also *Figure 2—source data 1*.

The online version of this article includes the following source data for figure 2:

**Source data 1.** The 593 statistically differentially expressed genes (false discovery rate [FDR] < 0.05) with at least 1.2-fold-change (FC ≥ 1.2) in severe preeclampsia (sPE) vs. control cases obtained from RNA-seq analysis.

## Comparison of DD transcriptomics in previous sPE in vivo vs. in vitro

We previously described DD in human endometrial stromal cells (hESCs) isolated from women with previous sPE compared to women with normal obstetric outcomes, but this finding was restricted to the stromal cell population using an in vitro decidualization cell culture model (*Garrido-Gomez et al., 2017*). Here, we compared DD overlapping between DEGs reported in vitro (n = 129) vs.

in vivo (n = 593) in sPE compared to control women. Nine genes were overlapped between the two datasets (*Figure 2A*); one gene was upregulated (*ERP27*), and eight genes were downregulated (e.g., *ISM1, MEST, MFAP2,* and *REEP2*). The expression pattern of common genes is presented as a box plot using counts per million, corroborating significant differential expression between sPE and control (*Figure 2B*). Recently, in vivo transcriptomics of endometrium at single-cell resolution across the menstrual cycle were characterized (*Wang et al., 2020*). Transcriptome profiles of stromal fibroblasts from the late secretory phase allowed the identification of deregulated genes in sPE as associated to hESC. We found that 263 genes from the 593 DEGs in sPE vs. control are expressed by hESC (*Figure 2C*). Taken together, the in vivo assessment provides a broad spectrum of dysregulated transcripts comparing with previous in vitro findings, which includes a high concordance with in vivo hESC genes.

## Identification of the fingerprint encoding human endometrial DD

To formulate the transcriptomic signature that encodes DD detected in sPE in vivo, we selected genes with significant dysregulation (FDR < 0.05) and at least 1.4-fold increase (FC ≥1.4) between sPE and control with assigned EntrezID. A volcano plot shows 120 DEGs meeting these criteria included in the final DD signature (*Figure 3A*; complete list of genes is included in *Figure 3—source data 1*).

GO analysis of the gene signature associated with DD in sPE identified 151 enriched biological processes downregulated (FDR < 0.05). These pathways were associated with cell cycle, DNA damage response, cell signaling, cellular response, cell motility, extracellular matrix, immune response, and reproductive process (*Figure 3B*). All are hallmarks of impaired decidualization and sPE pathogenesis. We identified fingerprinting genes representative of the altered pathways in sPE, such as *IL6* and *TNF*, regulating the response to bacterial molecules, *MMP3* and *MMP1* participating in the extracellular matrix organization, and *TNF*, *IL8*, and *FGF1* implicated in the downregulated receptor signaling (*Figure 3C*). Functional analysis revealed that the 120 DEGs included in DD fingerprinting are implicated in pathways related to decidualization, corroborating the maternal contribution to sPE. Interestingly, the number of downregulated genes was higher than the number of upregulated genes in sPE compared to controls, suggesting that, in vivo, DD may be induced by the lack of expression of a subset of genes.

Based on the 120 genes included in the DD signature, PCA showed that sPE and control samples clustered separately in two groups, except for three control samples (C20, C21, and C22) (*Figure 4A*). High variance between groups was effectively captured in the first two principal components. Unsupervised hierarchical clustering analysis confirmed that gene fingerprinting effectively segregated the two groups: one encompassing mainly controls and the other mainly sPE samples (*Figure 4B*). The same three controls clustered with the sPE group, recreating the PCA results.

To validate the DD gene signature in an independent cohort of samples (sPE [n = 7] vs. control [n = 4]), PCA based on these transcripts effectively segregated samples in two homogeneous groups (*Figure 4C*), corroborated by hierarchical clustering (*Figure 4D*). These genes successfully grouped 100% of controls and 85.7% of sPE cases supporting DD in sPE.

## DD fingerprint in sPE is connected to ER1 and PR-B

Of the 120 genes in the DD signature, 94 endometrial enriched genes encode for specific proteins reported by the Human Protein Atlas (*Uhlén et al., 2015*). Interestingly, 45 of those genes (47.9%) were included in the transcriptome modulated by *ESR1* (*Hewitt et al., 2010*), and 43 genes (45.7%) overlapped with the transcriptome and cistrome associated with *PGR* (*Mazur et al., 2015*; *Figure 5A*). Regarding target genes of ER1 and PR, the database of Human Transcription Factor Targets (hTFtarget) reported 17 genes responsive to ER1 and 50 target genes modulated by PR, based on epigenomic, CHIP-seq, or motif evidence (*Zhang et al., 2020*).

We evaluated the interaction between steroid receptor signaling and the proteins encoded by DD fingerprinting genes in sPE by building a dynamic network including ER1 and PR. String software (*Jensen et al., 2009*) was used to construct network connections visualized with Cytoscape software (*Shannon et al., 2003*). The interactome contained 117 nodes directly interconnected by 361 edges (*Figure 5B*). This DD fingerprint network showed a highly enriched protein–protein interaction (PPI) in sPE; indeed, the interconnection between nodes was significantly higher than the 93 edges expected (PPI enrichment p<1.0e-16). Clustering revealed three main modules based on their connectivity

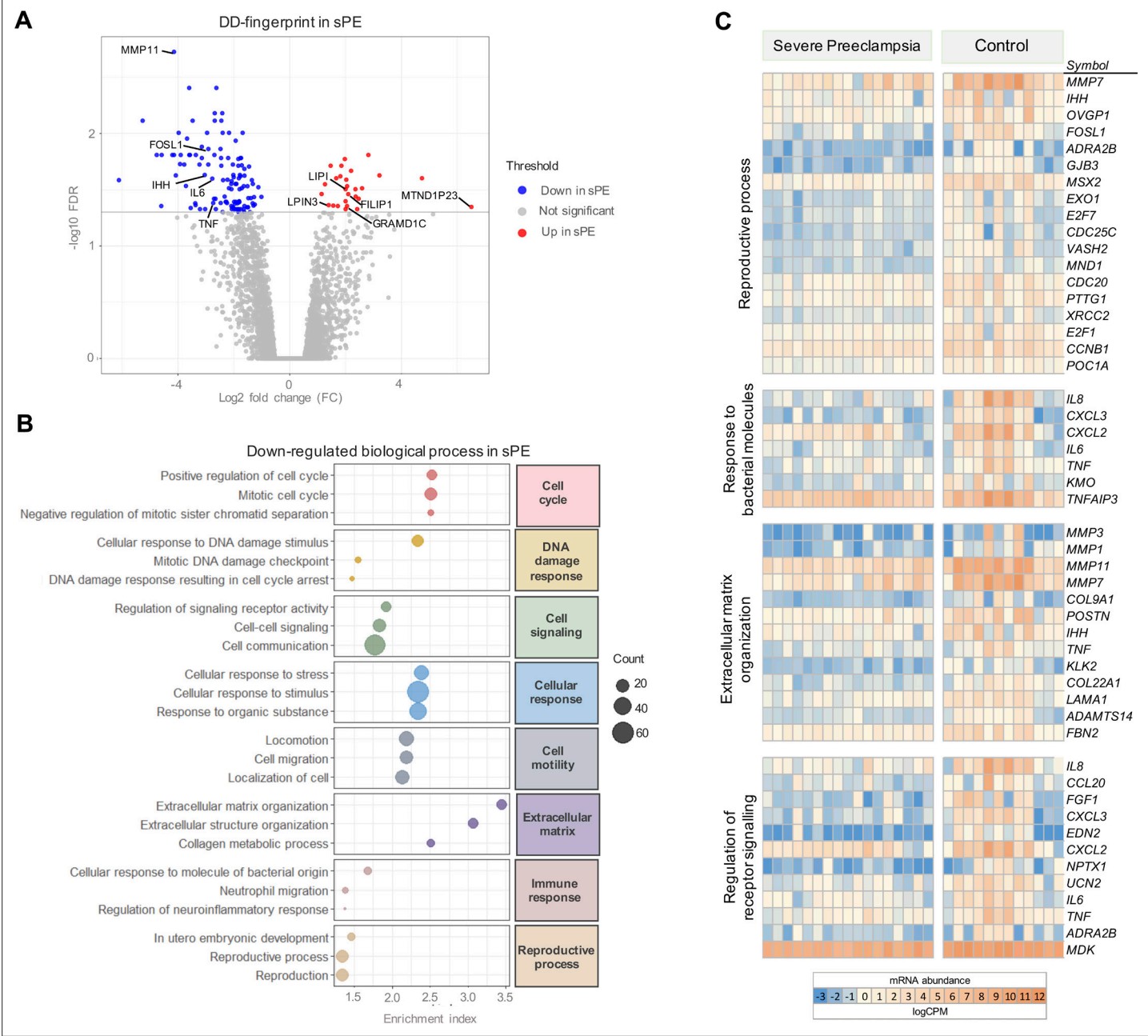

**Figure 3.** Severe preeclampsia defective decidualization (sPE-DD) fingerprint composed of 120 differentially expressed genes (DEGs). (**A**) Volcano plot showing downregulated (*blue*) and upregulated (*red*) genes in sPE from the DD fingerprint. Each point represents one gene; gray points are the rest of the genes obtained in the global RNA-seq analysis. (**B**) The three most highly downregulated biological process for each major category (*red*, cell cycle; *yellow*, DNA damage response; *green*, cell signaling; *blue*, cellular response; *gray*, cell motility; *purple*, extracellular matrix; *pink*, immune response; *brown*, reproductive process). Enrichment index was calculated by -log(p-value). (**C**) Clustering of DD fingerprint genes shown for reproductive process, response to bacterial molecules, extracellular matrix organization, regulation of receptor signaling, and response to hormones. See also *Figure 3—source data 1* and *Figure 3—source data 2*.

The online version of this article includes the following figure supplement(s) for figure 3:

**Source data 1.** List of genes selected as defective decidualization signature in severe preeclampsia (sPE) (120 differentially expressed genes [DEGs] with false discovery rate [FDR] < 0.05 and fold-change [FC] ≥ 1.4).

**Source data 2.** Biological process Gene Ontology (GO) terms computed by the 120 genes included in the defective decidualization (DD) fingerprinting in severe preeclampsia (sPE) (N, number of genes associated to GO term; DE, number of genes differentially expressed in this GO term; P.DE, unadjusted p-value; FDR, adjusted p-value).

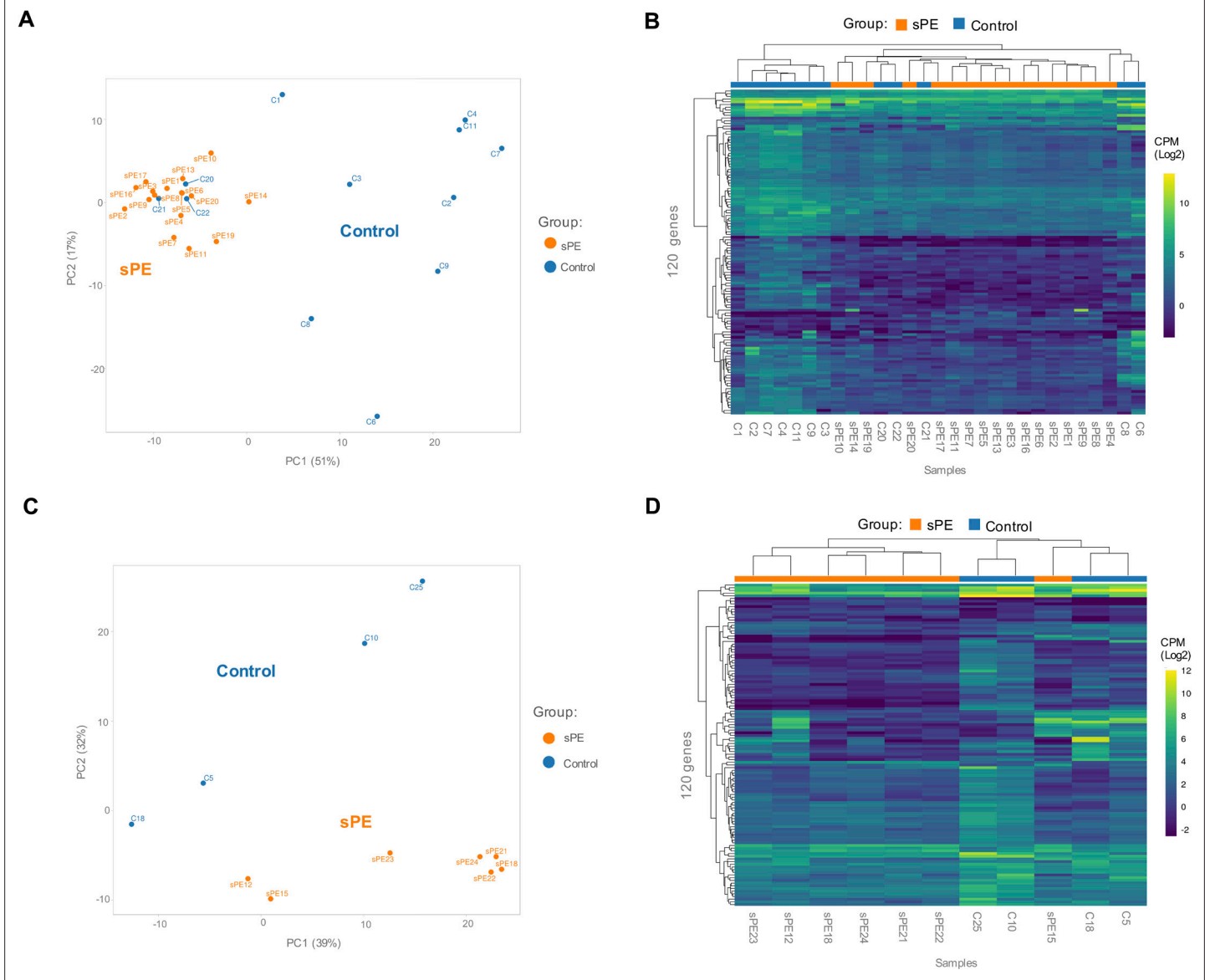

**Figure 4.** Validation of the defective decidualization (DD) fingerprint in severe preeclampsia (sPE). (**A**) Principal component analysis (PCA) based on 120 genes included in the fingerprinting in the training set. Each sample is represented as a colored point (*blue*, control; *orange*, sPE). (**B**) Heatmap dendrogram of expression of the 120 genes included in the final fingerprinting for each sample of the training set (control, n = 12; sPE, n = 17). (**C**) PCA based on the fingerprinting in the test set. Each sample is represented as a colored point (*blue*, control; *orange*, sPE). (**D**) Heatmap dendrogram of expression of the 120 genes included in the final fingerprinting for each sample of the test set (control, n = 4; sPE, n = 7). See also *Figure 4—source data 1*.

The online version of this article includes the following figure supplement(s) for figure 4:

**Source data 1.** List of genes selected as defective decidualization signature in severe preeclampsia (sPE) (120 differentially expressed genes [DEGs] with false discovery rate [FDR] < 0.05 and fold-change [FC] ≥ 1.4).

degree, with functionally relevant genes involved in gland morphogenesis, cell migration, extracellular matrix organization, stromal cell differentiation, cellular response to DNA damage stimulus, and regulation of cell cycle. The hub genes were determined by overlapping the top 10 genes obtained using two topological analysis methods in the cytoHubba plugin (*Chin et al., 2014*), MCC, and MNC. Five genes were selected, all of which were downregulated. Interestingly, both ER1 and PR were strongly embedded in the network and highly connected with DD fingerprinting, highlighting the interaction of hormonal receptors with notable decidualization mediators such as *IHH* and *MSX2* validated by RT-qPCR (*Figure 5C and D*). Furthermore, the interactome demonstrated a direct interaction between

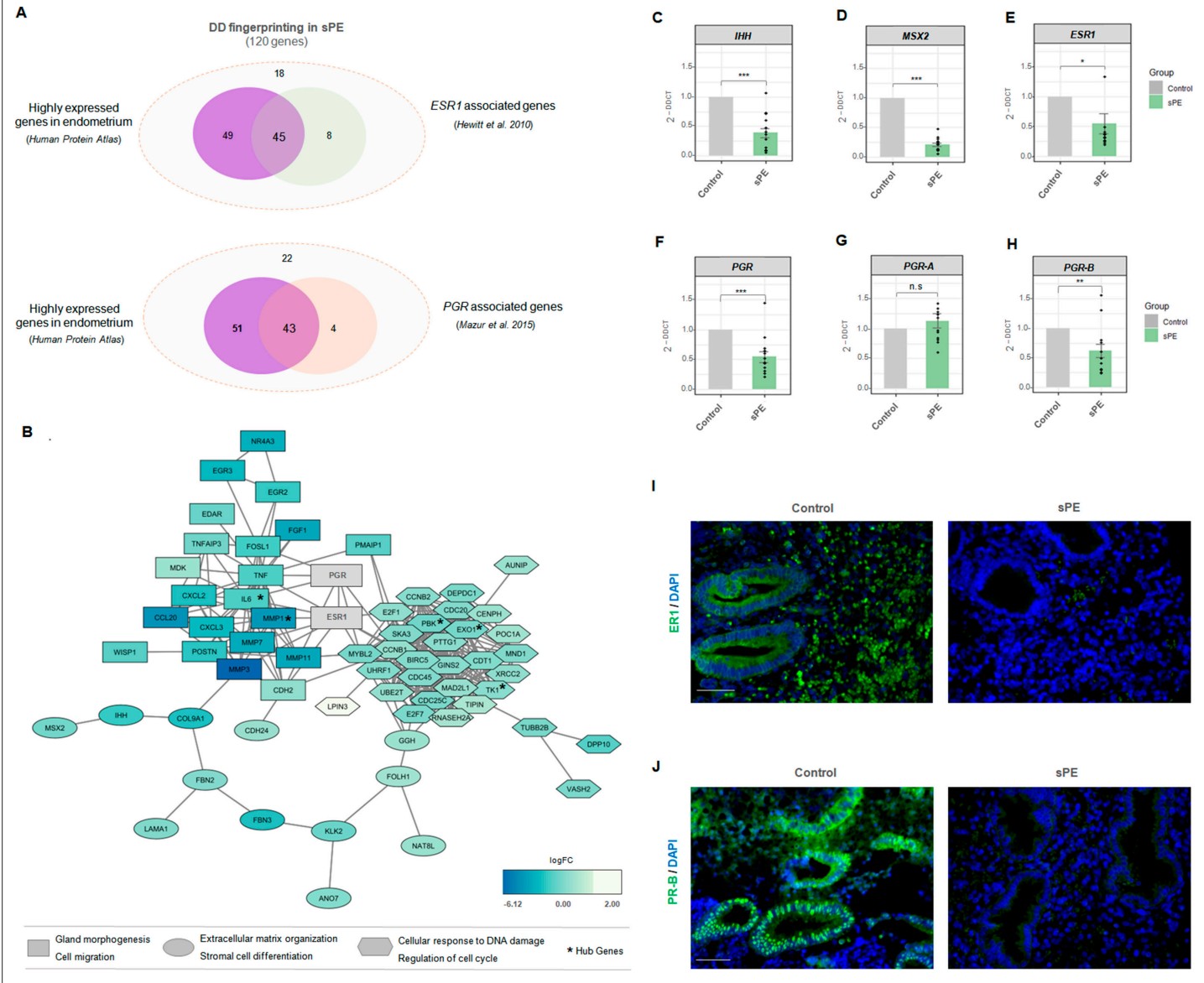

**Figure 5.** Estrogen receptor 1 (ER1) and progesterone receptor-B (PR-B) are linked to defective decidualization (DD) fingerprinting in severe preeclampsia (sPE). (**A**) Venn diagram displaying genes included in the fingerprinting (120) predominantly expressed in the endometrium based on Human Protein Atlas data that overlap with genes modulated by *ESR1* described by **Hewitt et al., 2010** and genes associated with *PGR* silencing described by **Mazur et al., 2015**. (**B**) Network showing the connections between proteins codified by DD fingerprinting and the hormonal receptors, ER1 and PR. Shapes indicate different clusters established by String k-means method. *Squares*, cluster involved in gland morphogenesis and cell migration; *circles*, cluster involved in extracellular matrix organization and stromal cell differentiation; *hexagons*; cluster involved in cellular response to DNA damage and regulation of cell cycle. Color gradient indicate gene expression in terms of log2FC. Hub genes are shown with an asterisk. (**C-H**) Gene expression levels of *IHH, MSX2, ESR1, PGR, PGR-A*, and *PGR-B* assessed for sPE (n=13) vs. controls (n=9) by RT-qPCR (*gray bars*, control; *green bars*, sPE). RT-qPCR values are expressed as mean± SE. *** p<0.001, ** p<0.01, *p<0.05. (**I-J**) Tissue sections of control (n=4) and sPE (n=4) endometrium during late secretory phase were immunostained with antibody against ER1 or PR. Nuclei were visualized with DAPI. Scale bar: 50 µM.

ER1 and PR. These results support the transcriptomic dysfunction of the genes present in the DD signature through imbalanced hormone receptor signaling in sPE.

We then analyzed the expression of *ESR1* and *PGR* in the endometrial tissue from a subset of women with prior sPE (N = 13) compared to controls (N = 9) by RT-qPCR. We found reduced expression of transcripts encoding the hormone receptors *ESR1* (p<0.05) and *PGR* (p<0.001) in sPE patients (*Figure 5E and F*). In-depth expression analyses revealed that the isoform *PGR-B* was significantly downregulated in sPE vs. controls (p<0.01), while the isoform *PGR-A* was unaffected (p>0.05)

(*Figure 5G and H*). These results were confirmed at the protein level by immunohistochemical analysis of ER1 and PR-B in endometrial biopsies collected in the late secretory phase from women with previous sPE (n = 4) and controls (n = 4) (*Figure 5I and J*). Both receptors were highly expressed through the decidualized endometrium, especially in the secretory glands in controls. In contrast, their expression was greatly reduced or absent in sPE samples. These results suggest that the DD transcriptomic signature implicates dysregulated ER1 and PR-B signaling in the late secretory phase in sPE patients.

## Discussion

Most scientific and clinical diagnostic efforts in sPE focus on placental surrogates. In this context, shallow cytotrophoblast invasion induces deficient vascular remodeling and ultimately aberrant placentation. This leads to placental ischemia and the release of soluble factors that induce the maternal syndrome, including the imbalanced levels of soluble fms-like tyrosine kinase 1 (sFLT1) and placental growth factor (PlGF) (*Rana et al., 2019*; *Powe et al., 2011*). sFLT1 protein binds to PlGF, preventing its interaction with endothelial receptors and leading to endothelial dysfunction. Accordingly, sFLT1 is increased, while free PlGF is decreased in serum from women with PE (*Levine et al., 2004*). The sFLT1/PlGF ratio has been proposed as a biomarker showing a positive predictive value of 36.7%, with 66.2% sensitivity and 83.1% specificity but its application is effective only 4 weeks before PE symptoms manifest (*Zeisler et al., 2016*). Therefore, during the first trimester there is a lack of high sensitivity and specificity screening methods to detect sPE early and prevent mortality and morbidity.

Current strategies based on placental dysfunction provide delayed results for preventive interventions and new approaches are urgently needed. Recent studies suggest that sPE might also be a disorder of the decidua, opening new avenues. In this sense, there may be a molecular signature associated with impaired endometrial maturation during early pregnancy in women who develop sPE (*Rabaglino et al., 2015*). Our previous work demonstrated that hESC isolated from patients with previous sPE failed to decidualize in vitro, suggesting a role of the maternal factor in the development of this disease (*Garrido-Gomez et al., 2017*). In addition, molecular pathways of dysregulated decidualization in PE are also found in endometrial disorders such as implantation failure, recurrent miscarriage, and endometriosis (*Rabaglino and Conrad, 2019*). Increasing evidence supports the detriment of inappropriate decidualization before pregnancy to reproductive outcomes, and the role of DD in the origin of sPE (*Garrido-Gómez et al., 2020*; *Ng et al., 2020*).

In the present study, we highlighted the underlying molecular defect that may explain in vivo decidualization failure as an important contributor to shallow placental invasion in sPE. For this purpose, we investigated the role of the decidua by leveraging global RNA-seq in late secretory endometrium. We identified 593 genes differentially expressed in sPE compared to controls, including genes involved in decidualization such as *PRL*, *IL6*, and *IHH*, as well as several novel genes. Nine of the DEGs overlapped with DEGs obtained in our previous in vitro decidualization study. In vivo endometrial biopsies include other cells in addition to hESC, and thus the expected degree of overlap between both in vitro and in vivo approaches should be modest as such was observed. However, we identified a large percentage of those DEGs that were associated with in vivo transcriptomic profile of hESC resolved at cell level from a healthy late secretory endometrium. Thus, the high number of our identified in vivo DEGs reflects the high complexity of decidualization at cellular heterogeneity in a physiological stage of women with previous sPE.

Previous reports of the decidual transcriptome in PE and sPE (*Garrido-Gomez et al., 2017*; *Løset et al., 2011*) revealed the gene expression profile associated with the condition at the time of delivery. Here, we analyzed samples collected years after the affected pregnancy; thus, it could be interesting to find dysregulated genes in common among these approaches. We compared our transcriptomic results and the previously reported dysregulated decidual genes, obtaining 1.3% (*Garrido-Gomez et al., 2017*) and 1.7% (*Løset et al., 2011*) overlap in affected genes. These discordances may not be unexpected and are consistent with recent results (*Rabaglino and Conrad, 2019*). The decidua basalis transcriptome at delivery was compared with the transcriptome of decidua at ~11.5 gestational weeks and in vitro decidualized hESC from women who experienced sPE. Both analyses revealed little, if any, overlap between molecular signatures. In contrast, the signature encoding in vitro DD of hESC years after pregnancy overlapped significantly with the dysregulated profile found in decidual samples at the beginning of pregnancy (*Rabaglino and Conrad, 2019*). Thus, decidual gene expression patterns

during the clinically active disease largely differ from those observed during endometrial decidualization at the end of the menstrual cycle and early pregnancy, perhaps reflecting a consequence rather than the origin of sPE.

From the 593 genes differentially expressed in sPE compared to controls, we identified a DD signature comprising 120 genes associated with sPE. This sPE-DD signature includes genes that allowed us to segregate samples from the training set into sPE and control groups, which were confirmed by the test set. One sPE was misclustered in the test set, such that 90.9% of samples from an independent cohort were properly clustered in the dendrogram. Having controlled for confounding effects of biological and technical variables, we consider that this misclustering is consistent with the nature of decidualization and its inherent variability. Decidualization is a highly dynamic process governed by (1) inter-individual variability in the endometrial menstrual cycle supported by displacement of the window of implantation in one out of four patients experiencing recurrent implantation failure (*Wang et al., 2020*; *Diaz-Gimeno et al., 2011*), (2) the physiology of the spatial expansion of decidualization process starting in some areas around spiral arteries and extending to the entire endometrium during the last days of the menstrual cycle (*Gellersen and Brosens, 2014*), (3) and the random spatial sampling during the endometrial biopsy that could influences cell-type proportions, which is inherent to the experimental strategy used in our investigation. Decidua sample segregation is consistent with the results presented by *Munchel et al., 2020*, who classified PE patients based on circulating RNA (C-RNA). In our study, we determined that the gene expression associated with DD had the highest potential to segregate sPE.

Interestingly, most of the DD fingerprint genes in sPE were related to the downregulation of *ESR1* and *PGR*, specifically *PGR-B*. We hypothesize that low expression of *PGR-B* and *ESR1* activates endometrial decidualization by dysregulating progesterone (P4) and estrogen (E2) action. Consequently, P4- and E2-related cellular signaling may be compromised, leading to the development of DD. Stromal cell differentiation, stromal–epithelial crosstalk (*Wang et al., 2017*), extracellular matrix degradation (*Itoh et al., 2012*), immune system response, and endothelial function (*Okada et al., 2018*) may all be disrupted by low expression of *PGR-B* and *ESR1*. Remarkably, the local immune system appeared to be dysregulated due to the altered expression of interleukins, cytokines, chemokines, and immunoglobulin, which could disrupt the tolerant microenvironment at the maternal–fetal interface in pregnancy and contribute to the development of sPE (*Erlebacher, 2013*; *Harris et al., 2019*).

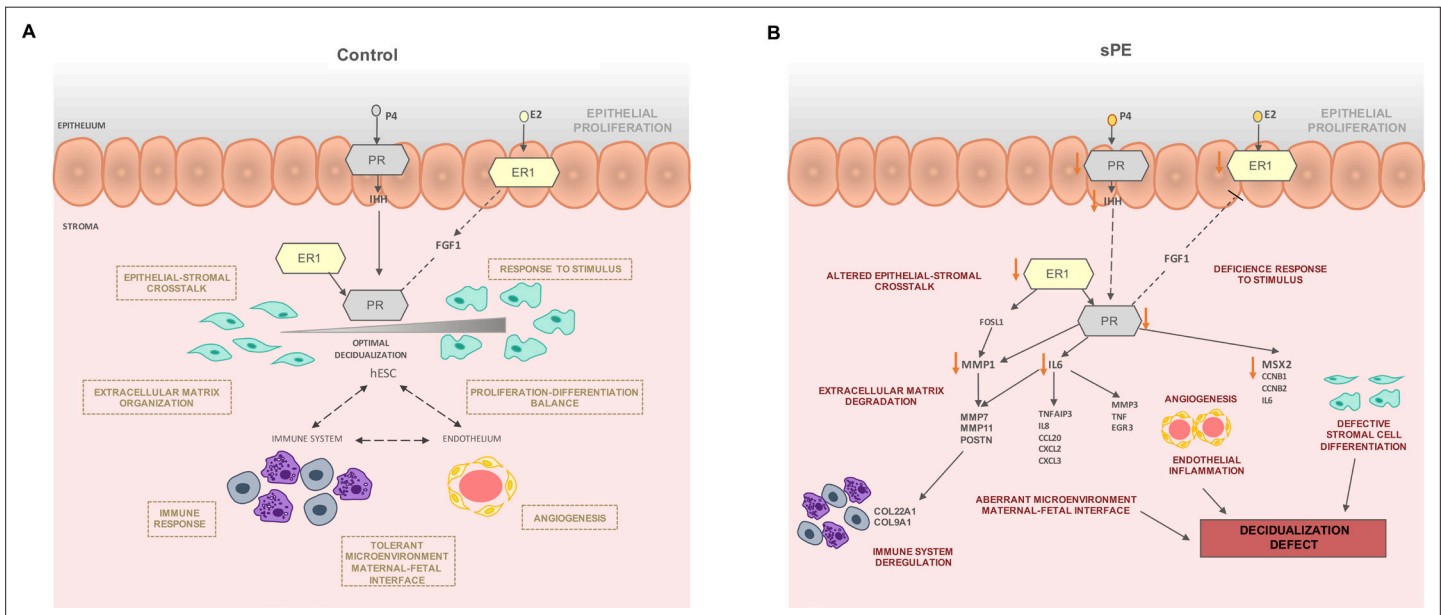

**Figure 6.** Modeling of the molecular mechanism for defective decidualization (DD) in severe preeclampsia (sPE). (**A**) Decidualization induced by P4 and E2 in control pregnancy including the interaction of immune response and endothelium. (**B**) Hypothetical network that could link DD and dysregulated hormone signaling in sPE. All genes were downregulated. Biological processes specified are candidates to be impaired based on functions associated with the observed dysregulation. *Red arrows* show the downregulation of decidualization modulators.

Based on our findings, we postulate that in non-sPE pregnancies balanced hormonal signaling leads to proper decidualization, which in turn interacts with immune and endothelial cells to control cytotrophoblast invasion (*Figure 6A*). P4 and E2 activate their receptors in the epithelium and signal to the stromal compartment. Likewise, stromal PR is activated by ER1, which induces target genes involved in decidualization such as *MSX2*, *CCNB1*, and *IL6*. In contrast, in sPE, endometrial *ESR1* and *IHH* are downregulated, decreasing *PGR* expression (*Figure 6B*) and leading to compromised decidualization, endothelial dysfunction, and local immune dysregulation.

In sPE, we found evidence of impaired epithelial–stromal crosstalk through *IHH* and *FGF1*, which could lead to an imbalance of hESC proliferation and differentiation (*Cha et al., 2012*; *Wang et al., 2018*). In support of this, we observed downregulated genes involved in stromal cell differentiation including *IL6, CCNB1, CCNB2*, and *MSX2*. Further, metalloproteinases such as *MMP1*, *MMP7*, and *MMP11* were downregulated in sPE, which could be associated to defective cell motility and extracellular matrix degradation. Other transcriptomic deregulations in sPE are connected to endothelial and immune system dysfunction, such as *EGR3* and *MMP3*, both of which are associated with altered angiogenesis (*Chen et al., 2020*; *Frieling et al., 2020*); *TNF* is involved in endothelial inflammation disbalance (*Marcos-Ramiro et al., 2014*) and influences the tolerant microenvironment at the maternal–fetal interface (*Li et al., 2018*). Taken together, our findings reveal a DD transcriptomic fingerprint in sPE driven by an imbalance in ER1 and PR signaling.

Our findings reveal significant gene expression dysregulation underlying DD in the late secretory phase in women who have had sPE. The potential origin of sPE may lie in the downregulation of *ESR1* and *PGR-B*. Both receptors are strongly coordinated to regulate decidualization and *PGR* expression is induced by ER1, which is inhibited by PR (*Patel et al., 2015*). E2 and P4 act through ER1 and PR in the epithelium and stroma and modulate the transcriptome and to promote crosstalk between both compartments (*Winuthayanon et al., 2010*). ER1 in the epithelium regulates stromal decidualization via paracrine mechanisms mediated by leukemia inhibitory factor (LIF), which controls *IHH* expression that transduce the signal activation of PR in the stroma (*Pawar et al., 2015*). Also, stromal ER1 activates PR in the same compartment (*Kaya Okur et al., 2016*). These findings could inform the development of therapeutic targets to restore optimal decidualization in sPE. Further studies are needed to fully elucidate the hormone signaling pathways that become dysregulated in sPE and the role of DD in the manifestation of the condition.

Preeclampsia is a syndrome in which different condition types might coexist; here, we focused on the maternal contribution to sPE through DD. We are not claiming that our findings are related to sPE heritability (DNA), but rather to the pathogenesis of the decidualization defect (RNA expression) involved in the origin of PE (*Garrido-Gómez et al., 2020*; *Ng et al., 2020*). Our findings reinforce a maternal cause for sPE through DD. This condition may result from an aberrant response to estrogen and progesterone mediated by ER1 and PR-B signaling. However, the primary driver of the predisposition to undergo decidualization resistance and its link with the main risk factors of sPE remain to be determined. Our work is an important step toward the development of new strategies that enable early assessment of risk for sPE and might prompt new therapeutic strategies to treat this enigmatic pathological condition. Future studies should focus on the translational potential of the DD fingerprinting to develop new noninvasive strategies based on circulating RNA to improve diagnosis and prognostication for women with sPE.

## Acknowledgements

We are grateful to Dr. Alfredo Perales and the Obstetric Area from the Hospital University and Polytechnic La Fe Hospital for invaluable help in the enrollment of participants and obtaining the tissue samples that made this study possible. We thank the University and Polytechnic La Fe Hospital recruiters Rogelio Monfort, Reyes Climent, Laura Rubert, Joana Dasí, and Julia Escrig for their assistance in compiling clinical data. We are indebted to the patient participants. This work was supported by the grant PI19/01659 (MCIU/AEI/FEDER, UE) from the Spanish Carlos III Institute awarded to TG-G. NC-M was supported by the PhD program FDGENT/2019/008 from the Spanish Generalitat Valenciana. IM-B was supported by the PhD program PRE2019-090770 and funding was provided by the grant RTI2018-094946-B-100 (MCIU/AEI/FEDER, UE) from the Spanish Ministry of Science and Innovation with CS as principal investigator. This work was funded partially by Igenomix S.L.

## Additional information

### Funding

| Funder | Grant reference number | Author |
| --- | --- | --- |
| Carlos III Health Institute | Grant PI19/01659 (MCIU/AEI/FEDER, UE) | Tamara Garrido-Gomez |
| Spanish Generalitat Valenciana | PhD Student grant FDGENT/2019/008 | Nerea Castillo-Marco |
| Spanish Ministry of Science and Innovation | PhD Student grant PRE2019-090770 | Irene Muñoz-Blat |
| Spanish Ministry of Science and Innovation | Grant RTI2018-094946-B-100 (MCIU/AEI/FEDER, UE) | Carlos Simon |

The funders had no role in study design, data collection and interpretation, or the decision to submit the work for publication.

### Author contributions

Tamara Garrido-Gomez, Conceptualization, Funding acquisition, Investigation, Methodology, Project administration, Resources, Validation, Visualization, Writing – original draft, Writing – review and editing; Nerea Castillo-Marco, Data curation, Formal analysis, Investigation, Methodology, Validation, Writing – original draft; Mónica Clemente-Ciscar, Data curation, Formal analysis, Investigation, Writing – original draft; Teresa Cordero, Investigation, Methodology, Project administration, Validation; Irene Muñoz-Blat, Investigation, Methodology, Writing – original draft; Alicia Amadoz, Data curation, Formal analysis, Methodology; Jorge Jimenez-Almazan, Data curation, Formal analysis, Investigation; Rogelio Monfort-Ortiz, Reyes Climent, Investigation, Project administration, Resources; Alfredo Perales-Marin, Resources, Supervision, Writing – review and editing; Carlos Simon, Conceptualization, Funding acquisition, Investigation, Project administration, Resources, Supervision, Writing – review and editing

### Author ORCIDs

Tamara Garrido-Gomez (iD) http://orcid.org/0000-0002-6584-4832
Nerea Castillo-Marco (iD) http://orcid.org/0000-0002-4817-4777
Alicia Amadoz (iD) http://orcid.org/0000-0003-3915-0404
Rogelio Monfort-Ortiz (iD) http://orcid.org/0000-0001-7931-8609
Alfredo Perales-Marin (iD) http://orcid.org/0000-0002-2221-2560

### Ethics

Human subjects: This study was approved by the Clinical Research Ethics Committee of Hospital La Fe (Valencia, Spain) (2011/0383), and written informed consent was obtained from all participants before tissue collection and all samples were anonymized (Included in Methods section - Human donors).

### Decision letter and Author response

Decision letter https://doi.org/10.7554/eLife.70753.sa1
Author response https://doi.org/10.7554/eLife.70753.sa2

## Additional files

### Supplementary files

• Supplementary file 1. Maternal and neonatal characteristics of endometrial donors.

• Supplementary file 2. Biological and technical variables of interest for controlling confounding effects in the RNA-seq analysis.

• Supplementary file 3. RT-qPCR primers list.

• Transparent reporting form

## Data availability

Transcriptomic data were deposited in the Gene Expression Omnibus database (accession number GSE172381) (Included in results and material and methods section). Custom scripts are available on GitHub at link https://github.com/mclemente-igenomix/garrido_et_al_2021, copy archived at https://archive.softwareheritage.org/swh:1:rev:c5a81d119229bc9d46c127c039a85de327f9b9c5.

The following dataset was generated:

| Author(s) | Year | Dataset title | Dataset URL | Database and Identifier |
|---|---|---|---|---|
| Garrido-Gomez T, Castillo-Marco N, Clemente-Ciscar M, Cordero T, Muñoz-Blat I, Amadoz A, Jimenez-Almazan J, Monfort R, Perales A, Simón C | 2021 | Disrupted PGR-B and ESR1 signaling underlies defective decidualization linked to severe preeclampsia | https://www.ncbi.nlm.nih.gov/geo/query/acc.cgi?acc=GSE172381 | NCBI Gene Expression Omnibus, GSE172381 |

The following previously published datasets were used:

| Author(s) | Year | Dataset title | Dataset URL | Database and Identifier |
|---|---|---|---|---|
| Garrido-Gomez T, Dominguez F, Quiñonero A, Diaz-Gimeno P, Kapidzic M, Gromley M, Ona K, Padilla-Iserte P, McMaster M, Genbacev O, Perales A, Fisher SJ, Simón C | 2017 | Defective Decidualization During and After Severe Preeclampsia | https://www.ncbi.nlm.nih.gov/geo/query/acc.cgi?acc=GSE94644 | NCBI Gene Expression Omnibus, GSE94644 |
| Wang W, Vilella F, Alama P, Moreno I, Mignardi M, Isakova A, Wenying P, Simon C, Quake SR | 2020 | Single cell RNA-seq analysis on human endometrium across the natural menstrual cycle | https://www.ncbi.nlm.nih.gov/geo/query/acc.cgi?acc=GSE111976 | NCBI Gene Expression Omnibus, GSE111976 |
| Hewitt SC, Kissling GE, Fieselman KE, Jayes FL, Gerrish KE, Korach KS | 2010 | Estrogen response uterine gene profile in Ex3 $\alpha$ ERKO | https://www.ncbi.nlm.nih.gov/geo/query/acc.cgi?acc=GSE23072 | NCBI Gene Expression Omnibus, GSE23072 |

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
