## [Decision Letter]

**Acceptance summary:**

Garrido-Gomez et al., generate a unique dataset profiling the transcriptomes late-stage endometrium of women with prior pregnancies leading to severe preeclampsia (sPE) compared to that of non-preeclamptic pregnancies. Although the question of molecular drivers preeclampsia has been explored at the molecular level directly in the placenta, this study provides a unique view in the endometrium of women years after the pregnancy in question, which is consistent with the fact that a previous preeclamptic pregnancy is one of the best predictors for a subsequent preeclamptic pregnancy.

**Decision letter after peer review:**

Thank you for submitting your article "Disrupted PGR-B and ESR1 signaling underlies preconceptional defective decidualization linked to severe preeclampsia" for consideration by *eLife*. Your article has been reviewed by 3 peer reviewers, one of whom is a member of our Board of Reviewing Editors, and the evaluation has been overseen by Kathryn Cheah as the Senior Editor. The reviewers have opted to remain anonymous.

Essential revisions:

Although the 3 reviewers noted the potential of the study, major concerns were raised about the soundness/clarity of the transcriptomics analysis, as well as lack of clarity on the study population. Specifically:

1. The manuscript needs to undergo major revisions in the statistics and transcriptomics analyses (see specific points with details in reviewers 1 and 2 comments):

a. The use of fold-change thresholding needs to be removed/updated.

b. Multiple hypothesis correction needs to be used prior to determining significance with genome-wide datasets. Please use FDR correction or other accepted methods.

c. Only use Student t-tests when data normality can be confirmed, otherwise, use non-parametric tests.

d. Since this study is done on endometria sampled *years* after actual pregnancy, it would be crucial to compare to public datasets on actual placentas, to determine overlap and differences.

e. PCA cannot be used to determine clustering or lack thereof, as well as whether differences exist or not.

f. There are concerns about batches and co-variates not taken into account (see reviewers 1 and 3 points) which could lead to the absence of DEGs. These need to be clearly stated and provided in a table. They also need to be corrected for, either using batch removal and/or multivariate modeling.

g. A continuous GSEA-type analysis would be more informative than a thresholded approach for functional enrichment (i.e. GO) analysis.

h. RT-qPCR (especially on the exact same sample set) is not an appropriate validation for robust RNA-seq datasets.

2. The authors needs to substantially revise and improve the details on methodology (i.e. RIN threshold), as well as deposit all code and data to public databases for re-analysis. This is crucial for long-term reproducibility and use of this as a resource. Analytical choices against the general rule of the field (such as using hg19 and not the more recent better annotated hg38) should be justified or at least explicitly addressed.

3. The patient population needs to be better defined (see reviewer 3 points, and to a lesser degree reviewer 1).

4. Make sure to define all abbreviations, and avoid using non-standard abbreviations to improve readability.

*Reviewer #1 (Recommendations for the authors):*

1. There are major concerns about the statistics in the analysis of the dataset.

a. Fold-change thresholding is used in multiple places in the analysis (e.g. line 94, 112, 128, etc.). This is a big problem, as it is known to lead to poor FDR control in the case of statistical tests not designed to take fold change into account (like edgeR used here). Thus, the practice of fold-change filtering without FDR control is considered problematic by statisticians (see PMID: 19176553). Thus, either the authors need to update their results after removing this problematic criteria to properly control for FDR, or all analyses should be rerun with a method that takes fold-change into account for FDR control (e.g. the TREAT method PMID: 19176553).

b. For the GO enrichment analysis (line 133), the authors only indicate a p-value filter (p < 0.005). Was there not implementation of multiple testing correction? The data should be re-analyzed after proper multiple hypothesis correction control (native DEseq2 FDR, Bonferroni, or other). In addition, what was the gene list background used to compute enrichment? The nature of the background has been shown to hugely impact results and needs to be explicitly detailed in the methods.

c. The authors reports using Student t-tests (section starting line 404). Unless normality is proven (e.g. with a Shapiro-Wilkes test), parametric tests that relie on normal distributions should never be used. We recommend that all tests be rerun with non-parametric equivalents (e.g. Wilcoxon or Mann Whitney) to guarantee these assumptions are not violated.

2. With human sample studies, it is unlikely that all samples were processed exactly in parallel (i.e. all samples collected the same day, kept in stabilizer for the same amount of time, etc.), which all but guarantees the existence of batch effects. There are also a number of stated parameters (i.e. age of the donor, time since pregnancy 1-8 years, day in cycle 22-32, etc.) that could influence results without a relevant biological driver.

a. In addition to an aggregated table (Table 1), the authors should generate a table for each anonymized sample for key characteristics so that the resource can be analyzed with these caveats considered both in this current paper and to facilitate all future reanalysis attempts. This should include for each sample individually:

i. age of the donor.

ii. time since last pregnancy.

iii. day of cycle for biopsy.

iv. date of processing/batch (including biopsy batch, library batch AND sequencing batch).

v. RIN of RNA sample.

vi. Sequencing depth (raw reads per library).

vii. Percent reads mapping to the genome reference (here hg19).

viii. Ethnicity of donor.

b. The authors should implement a multivariate model to account for all these potential confounding effects so as to focus on the core biological signal. In addition, the use of a software like SVA or RUVseq is recommended to remove unwanted technical variation prior to analyses.

3. Additional methodological information is needed for long-term reproducibility of analyses.

a. For reproducibility of code and analyses, all analytical code for this study should be deposited in a repository such as Github or made available as a Supplementary file.

b. Please include all version numbers for all used software (e.g. R, etc.) packages and R packages (e.g. edgeR, goana, etc.), as well as all command parameters where relevant. The same should be applied to annotation databases (i.e. GO used here), and if a version number doesn't exist, date of access should be provided.

c. There is no mention of the softwares (nor versions) used to trim RNA-seq reads, to map to the hg19 genome or to aggregate counts to genes. All used software should be clearly named and their use described with all necessary options for reproducibility.

d. Line 84-88/Figure 1A, were the subtypes of "control" samples (pre vs. full-term) equally distributed in training vs. testing? Indeed, although the authors argue that the subtype made no difference on transcriptional profiles (i.e. Sup Figure 1A PCA), this reviews sees a general (if not clean separation) on PC1, that would probably be enhanced if sources of spurious variation (see point #2) were taken into account. In any case, the data provided cannot be used to support the statement "demonstrated an absence of clustering" (line 80), since the statistical tests used are not meant to prove the null hypothesis but to reject it when possible (i.e. confusion between type I and II error). Please amend the analysis to reflect this.

e. cDNA library cannot be validated for RIN (line 330), as only total RNA samples can be. The entire method section needs to be completely rechecked and rewritten for accuracy to avoid this kind of errors.

f. All used QC filters should be stated explicitly in the methods (see lines 338-342).

4. Since RT-qpCR is known to be less sensitive and more prone to normalizing biases (due to the choice of control genes which may vary themselves) than RNA-seq (which performs unbiased normalization at the transcriptome-wide level), it is unclear why validation was performed on the same samples that were processed by RNA-seq (line 375-389). Unless the authors want to include RT-qPCR data on an INDEPENDENT cohort of patients, these results are circular and shouldn't be included in a revised manuscript. If RT-qPCR on a new cohort is included, make sure to include information on which normalizing amplicons were used for δ CT calculation.

5. Can the authors explain why 3 specific samples (C20, 21, 22; line 147-152) do NOT cluster appropriately with their "signature"? Did they have any biological specificities (see need for sample-by-sample information as raised in point #2)? The fact that 3 control samples cluster with the sPE samples somewhat invalidate the signature as a signature of sPE, and may be the result of improper correction for batching or other technical (or irrelevant biological) noise. Please discuss explicitly what may be happening here in a revised manuscript.

*Reviewer #2 (Recommendations for the authors):*

In sum, I feel that this paper brings some interesting insights on the decidua status and preeclampsia; the degree of novelty is nevertheless unclear to me, and the clinical application seems far-fetched. There are several points that may be better presented and discussed.

Detail of the recommendations to the authors:

1. The study of the decidual transcriptome in preeclampsia has also already been performed by other teams in the past, rather at the moment of the disease than later. For instance, the paper of Mari Loset (Am J Obst Gynecol, 2011), is, I feel, an important base for comparison with the results presented here. So much so, that it used the same type of expressional approaches, and detected differential transcripts and cascades. I feel that the authors should analyze their own datasets in light of this type of seminal papers and discuss the commonalities and differences found.

2. In the overrepresented pathways in the Loset's work, the regulation by ESR1 and PGR was not obvious. Does it mean that analyzing samples 3-4 years after the event (normal or pathological pregnancy) leads to remnants of the disease at the uterine level, or is there a genetic predisposition? This is a question that cannot be truly addressed by the present dataset, and is important if the objective is to define markers, as stated below.

The introduction gives the necessary details to understand the question raised, finding a signature that could help assessing the risk of preeclampsia.

3. However, the genetic part of preeclampsia is estimated by a heritability of ~55%. The genetic decomposition of this heritability presented by Cnattingius and coworkers (2004) indicate that only part of this genetics is connected to the maternal genetic background (which concerns two organs: the uterus and thus the decidua) and the placenta for half of its genetics. Part of the genetics of preeclampsia is associated only to the paternal genetics (estimated at ~1/3 of the heritability), therefore, it is clear that the risk cannot systematically be found by analysis of the female (uterine-decidua) expression profile only. Alternatively, the difference observed by the authors may be a consequence of the previous preeclampsia not based on a predisposition, but if this is the case, it limits considerably the usefulness of the markers found since preeclampsia, and in particular severe preeclampsia is at 75% a disease of the first pregnancy. The authors should recognize this as an important limit as an early prenatal screening strategy.

The initial analysis is based upon the endometrial tissue of 24 women that had a severe preeclampsia, 16 controls (8 preterm and 8 term births). Gestational age at delivery had apparently no influence on the 'control' groups. Since the samples were not collected at the end of gestation, which means that the occurrence of preterm may be connected to the placenta and not to the maternal uterine situation.

4. Nevertheless, there are only 8+8 samples so it is not possible to prove that in some case a decidual expression is not involved. In addition, using PCA to check that there is no clustering is not enough to certify that there are no differentially expressed genes that separate the two groups, it means only that the number of differential genes compared to the mass of genes that are not differentially expressed is not enough to cluster the groups, which may be the case if the percentage of genes changed is small. There are some published evidences that recurrent spontaneous abortion, pre-term birth and preeclampsia share common mechanisms. The observation here seems to indicate that the uterus of women with PTB is identical to the one of term pregnancies. From the clinical data, the authors acted wisely in taking PTB without hypertension. But it is not clear from the data that the women that had PTB had a recurrent occurrence in this (not mentioned in Table S1).

5. In addition, the absence of DEG genes visible (Supplementary Figure 1B) is presented against FDR and not p-value which is unusual and I think a bot too stringent. Even random samples should give some significant genes (one out of 20) Using FDR leads to no significant genes, which is not a complete surprise. However, it does not mean that individual genes are not relevant for the difference between the two situations analyzed. Another interesting approach should be to check for enrichment of pathways using GSEA approaches, that do not rest on the establishment of thresholds that are always arbitrary. In sum excluding totally the existence of DEG between the two groups seem a bit rapid.

6. In the legend of Suppl figure S1, I do not understand 'Plot based on 728 genes labeled as fc (blue) and 17,748 genes labeled as none (purple)'. Nevertheless, it is admissible that the differences due to preterm or term are negligible compared to the differences induced by sPE.

The decomposition of the samples between sets for simulations is unusual in the field, but mathematically sound. The authors discovered 859 DEG at a threshold of 2-fold, and some genes (9) were validated by qRT-PCR.

7. It would be nice to analyze the data not only relative to a threshold of induction, but taking the complete dataset and using GSEA-like approaches to see whether they are consistent with the gene clusters found by threshold-volcano plots analyses.

Previous studies by the authors evaluate gene expression differences in decidualization either from human endometrial stromal cells (hESCs) from women with sPE versus normal. The authors identified 18 genes differentially expressed in vivo and in vitro in sPE compared to control.

8. The authors do question the observation of having only 18 common genes between in vivo and in vitro, out of 129 or 859 (in vitro – previous study- and in vivo -present study- respectively), and base them upon cell composition that is much more complex in vivo, which is reasonable. However, in the search for a signature, which is one of the justifications of the work, it could mean that the in vivo dataset could contain more 'relevant' genes compared to a in vitro model. The comparison with the results of Wang, which is much better (and focused better on a simple cell model) on the single-cell transcriptome may belong rather to a discussion rather than a results part of the paper. The validation by qRT-PCR is good, but the choice of the genes leads to a very high correlation (2D), that may be due to the use of only one induced gene (and only 5 genes used). About the actin as a control housekeeping gene, it is not sure that it is the best reporter gene in the uterine context. Generally, it is advised to normalize against the geometric mean of two to four different reporter genes.

A selection of 166 highly deregulated genes (>4 fold) is then selected, and were shown to be enough to separate efficiently the samples, which is not surprising, given that this corresponds to a semi-supervised analysis from genes found differential between the two gene sets.

9. The identification of Estrogen/progesterone receptor is not a surprise, when uterus function is concerned. I would suggest that the authors complete their analyses using network analyses such as provided by the combination of Stringdb and Cytoscape. This would help to visualize the pivotal position of ESR1 and PGR more clearly, or maybe to find other important hubs that were overlooked in their current study. Also using GSEA or other tools on transcription factor binding sites databases, showing the actual involvement (through measuring the enrichment, and calculation of FDRs) of the Estrogen Responsive Element and Progesterone Responsive Element would be an essential element that must be shown to prove a genuine enrichment in these cascades

*Reviewer #3 (Recommendations for the authors):*

There is much debate on whether the risk of PE is higher/lower in women with male vs. female fetuses. Additionally, the maternal response to pregnancy differs based on the sex of the fetus. The manuscript should therefore report on this variable and whether it impacts the results of the analysis.

Please clarify whether biopsies were done for clinical or research purposes only.

Why did the authors choose to use hg19 as their reference genome and not the better annotated hg38?

Methods (line 321) – What RIN value was defined as appropriate for making RNAseq libraries.

I cannot find anywhere where DD is defined. Can the authors please clearly define what they mean when they use DD?

Introduction – PE also significantly contributes to maternal mortality, not just infant.

Figure 3B – what is the axis label?

Figure 4 – I believe the authors mean sPE (not Spe).

Data availability: authors state that the RNAseq data is available for download in GEO.

[Editors' note: further revisions were suggested prior to acceptance, as described below.]

Thank you for resubmitting your work entitled "Disrupted PGR-B and ESR1 signaling underlies defective decidualization linked to severe preeclampsia" for further consideration by *eLife*. Your revised article has been evaluated by Kathryn Cheah (Senior Editor) and a Reviewing Editor.

The manuscript has been improved but there are some remaining issues that need to be addressed, as outlined below:

Essential revisions:

The authors failed to properly respond to key reviewer concerns, which were considered to be necessary changes by the consensus of reviewer's discussion. Thus, the manuscript cannot be further considered until these points raised by the reviewers are appropriately addressed.

1. The editor and reviewers understood the method used for fold change thresholding. No additional explanation was required. The method is still wrong, as post-FDR fold-change thresholding is the problem that was raised and cannot be used in a rigorous analysis. Thus, the authors should implement one of the previously given options so that this major problem is corrected: (i) get rid of fold-change thresholding completely, or (ii) use a method that does it, while appropriately controlling for FDR (e.g. TREAT).

2. Similarly, using multiple corrections for GO and KEGG analysis is not a suggestion but a necessity for rigorous analysis, especially when performing so many tests. If term redundancy is a major concern for the authors in the GO database, they are advised to look at the GOSlim framework which was put together to specifically respond to this concern. In any case, enrichment results without FDR control are non-reproducible and not statistically supported.

3. The authors have not responded to the requirement to upload all code/scripts generated for the study. They refer to the software repository (i.e. edgeR), but not the actual code for analysis. The specific scripts that were written to run all analyses need to be made available as well on GitHub or as a supplement, as per *eLife* policy.

---

## [Author Response]

Reviewer #1 (Recommendations for the authors):1. There are major concerns about the statistics in the analysis of the dataset.a. Fold-change thresholding is used in multiple places in the analysis (e.g. line 94, 112, 128, etc.). This is a big problem, as it is known to lead to poor FDR control in the case of statistical tests not designed to take fold change into account (like edgeR used here). Thus, the practice of fold-change filtering without FDR control is considered problematic by statisticians (see PMID: 19176553). Thus, either the authors need to update their results after removing this problematic criteria to properly control for FDR, or all analyses should be rerun with a method that takes fold-change into account for FDR control (e.g. the TREAT method PMID: 19176553).

We thank the reviewer for their feedback that allows us to explain in more detail the statistical analysis performed in the manuscript. First, we would like to highlight that all the analyses were performed with a proper false-discovery rate (FDR) cutoff as the basic criterion. Specifically, we used the exactTest function (edgeR package) that is recommended to make comparisons on datasets with a single-factor design, allowing us to achieve a global exploratory approach. This function uses the p-value adjustment method FDR, and a cutoff of 0.05 (FDR<0.05) was applied to identify significantly differentially expressed genes (DEGs). Once we obtained the DEGs, we selected those with a high fold change (FC≥2) to identify differences large enough to be biologically meaningful. We specified these two criteria (FDR<0.05 and FC≥2) in each line where the DEGs were described (e.g., Lines 94, 112, 128) and in the “Differential expression analysis” section in the Material and Methods (Lines 341–342), but we now provide more detail on the statistical analysis information used in this work.

– Specifically, we rewrote the “Quality control and pre-processing data” and “differential expression analysis” sections of the Materials and methods. These sections were replaced by a new section titled “RNA-seq analysis” (Lines 144-167): “Reads were mapped to the hg19 human genome transcriptome using the STAR (version 2.4.2a) read aligner (1). FastQC (version 0.11.2) was used to determine the quality of FASTQ files. The manipulation of SAM and BAM files was done with the software SAMtools (version 1.1) (2). To count the number of reads that could be assigned to each gene, we used HTSeq (version 0.6.1p1) (3) and BEDtools software (version 2.17.0) (4) to obtain gene coverage and work with bedFiles. Quality control filters at each program were used following the software package recommendations, and reads were filtered by mapping quality greater than 90%. Transcriptomic data were deposited in the Gene Expression Omnibus database (accession number GSE172381). The Bioconductor package edgeR (version 3.24.3) (5) was used to analyze differentially expressed genes. The trimmed mean of M-values normalization method was applied to our gene expression values. The exactTest function was used to find differentially expressed genes between groups. The p-value adjustment method was FDR with a cut-off of 0.05 (FDR<0.05). Once p-value was adjusted, significant deregulated genes with log2-fold-change ≥1 (FC≥2) were selected to perform gene ontology analysis and to formulate the signature encoding defective decidualization. edgeR analysis was carried out in R version 3.5.1. A volcano plot was created to visualize DEGs. For a better overview, we distinguished significant (FDR<0.05) and not significant (FDR≥0.05) DEGs with a high (FC≥2) or low fold-change (threshold FC<2).”

– We improved the explanation about the statistical analyses used in the Results section (Line 270-273): “Transcriptional analysis in the training set was performed by comparing gene expression patterns in sPE (n=17) and controls (n=12). This comparison revealed 859 significantly differentially expressed genes (DEGs) based on FDR<0.05 and at least two-fold differential expression between groups (FC≥2)” and (Line 293-295): “Eighteen genes were similarly differentially expressed with at least a two-fold change between groups (FDR<0.05 and FC≥2) (Figure 2A).”

– The legend of Figure 1B was also updated (Lines 673-675): “not significant-lowFC (FDR≥0.05, FC<2); not significant-highFC (FDR≥0.05; FC≥2); significant-lowFC (FDR<0.05; FC<2); and significant-highFC (FDR<0.05; FC≥2).”

– Supplementary file 3 has changed its name by “Figure1-Source data 1” and was included p-values, FDR values (corrected p-values), log FC, and FC, and we included how the criteria were applied in the legend, (lines 779-780) “Figure1-Source data 1. The 859 statistically differential expressed genes (FDR<0.05) with at least two-fold change (FC ≥ 2) in sPE vs control cases obtained from RNA-seq analysis.”

– The Supplementary file 4 have changed its name by “Figure 3-Source data 1” and legend was updated (783-784): “List of genes selected as the defective decidualization signature in sPE (166 DEGs with an FDR of <0.05 and an FC of ≥4).”

b. For the GO enrichment analysis (line 133), the authors only indicate a p-value filter (p < 0.005). Was there not implementation of multiple testing correction? The data should be re-analyzed after proper multiple hypothesis correction control (native DEseq2 FDR, Bonferroni, or other). In addition, what was the gene list background used to compute enrichment? The nature of the background has been shown to hugely impact results and needs to be explicitly detailed in the methods.

For this analysis, we followed the recommendation provided by the edgeR user’s guide:

“The p-values returned by goana and kegga are unadjusted for multiple testing. The authors have chosen not to correct automatically for multiple testing because GO terms and KEGG pathways are often overlapping, so standard methods of p-value adjustment may be very conservative. Users should be aware though that p-values are unadjusted, meaning that only very small p-values should be used for published results.”

For this reason, we selected a p-value filter of <0.005. Supplementary file 5 includes the p-value for each GO reported to provide statistical information to readers that are interested in using this information for their own research. Moreover, GO annotations emphasized in the text and in Figure 3 have a p-value of <0.005, with the vast majority lower than 0.00005. In addition, we agree with the reviewer that more information regarding the gene list background used to compute the analysis needs to be explicitly detailed, and we have added this information in Materials and methods.

– We rewrote the Enrichment analysis section of Materials and methods (Lines 180-188):

“GO analyses were conducted to obtain biological processes using the goana function in edgeR (62). The input genes were those 166 included in the fingerprinting (Figure 3-Source data 1) P-values returned automatically by the goana function are unadjusted for multiple testing because GO terms are often overlapping and standard methods of p-value adjustment may be very conservative. Thus, we kept GO terms with p-values of <0.005 (Figure3-Source Data 2); includes the p-value for each GO reported.”

– Supplementary file 5 have changed its name by “Figure 3-Source data 2”. The legend of was revised (785-787):

“Figure 3-Source data 2. Biological process GO terms computed by the 166 genes included in the DD fingerprint in sPE (N, number of genes associated with GO term; DE, number of genes differentially expressed in this GO term; P.DE, p-value).”

c. The authors reports using Student t-tests (section starting line 404). Unless normality is proven (e.g. with a Shapiro-Wilkes test), parametric tests that relie on normal distributions should never be used. We recommend that all tests be rerun with non-parametric equivalents (e.g. Wilcoxon or Mann Whitney) to guarantee these assumptions are not violated.

We followed the reviewer’s recommendation, and all the tests were rerun using the non-parametric method Wilcoxon to guarantee that assumptions of normality are not violated.

– The statistical analysis section of the Materials and methods was updated (Lines 232-234): “Clinical data were evaluated by Wilcoxon test for comparisons between sPE and control samples.”

– Supplementary file 1 was updated.

2. With human sample studies, it is unlikely that all samples were processed exactly in parallel (i.e. all samples collected the same day, kept in stabilizer for the same amount of time, etc.), which all but guarantees the existence of batch effects. There are also a number of stated parameters (i.e. age of the donor, time since pregnancy 1-8 years, day in cycle 22-32, etc.) that could influence results without a relevant biological driver.a. In addition to an aggregated table (Table 1), the authors should generate a table for each anonymized sample for key characteristics so that the resource can be analyzed with these caveats considered both in this current paper and to facilitate all future reanalysis attempts. This should include for each sample individually:i. age of the donor.ii. time since last pregnancy.iii. day of cycle for biopsy.iv. date of processing/batch (including biopsy batch, library batch AND sequencing batch).v. RIN of RNA sample.vi. Sequencing depth (raw reads per library).vii. Percent reads mapping to the genome reference (here hg19).viii. Ethnicity of donor.

According to the reviewer’s suggestion, we have generated a Supplementary file 2 that includes the key characteristics indicated about each anonymized sample. The impact of these variables was considered in our analysis following the reviewer’s recommendations.

New Supplementary file 2 has been included in the manuscript (Line 112-114).

b. The authors should implement a multivariate model to account for all these potential confounding effects so as to focus on the core biological signal. In addition, the use of a software like SVA or RUVseq is recommended to remove unwanted technical variation prior to analyses.

We agree with the reviewer about the relevance of potential confounding factors of key characteristics on our transcriptomics results. We have applied an experimental design to minimize the impact of the main sources of batch effect on our gene expression data. Endometrial biopsies were preserved with stabilization solution and frozen at −80º until analysis to obtain the best-quality RNA. We used a balanced batch-group design. Specifically, case and control samples were included and were balanced and processed at the same time in the same technical batch, including the RNA extraction, library preparation, and sequencing. The batches performed were reduced to the minimal number of three. Additionally, before conducting the transcriptomic analysis, the confounding effects of main technical and clinical variables were tested. Once confounding effects were discarded, we performed a differential gene expression analysis. We included in the revised manuscript this additional information used as a basic starting point in our data analysis. In (Author response image 1), we provide to the reviewer some data from analyses to control for confounding effects in our experimental design. To exclude confounding effects in the gene expression analysis, we applied a principal variance component analysis (PVCA) to fit a mixed linear model using biological and technical variables as random effects to estimate and partition the total variability.

**Author response image 1. sa2fig1:** 

The results demonstrated that the effect of biological and technical variables evaluated individually are less than 0.05, and their confounding effects are negligible. By contrast, the variable group (control and sPE) is the most plausible variable compared with the rest of the individual or double-interaction between variables (less than 0.1).– Materials and methods (Lines 128-129): “cDNA libraries from total RNA samples (n=40) were prepared using an Illumina TruSeq Stranded mRNA sample prep kit (Illumina, San Diego, CA) following a balanced batch-group design.”

– Results (Lines 247-248): “Biological and technical variables for each donor were considered to discard confounding effects on the transcriptomic profile (Supplementary file 2).”

– Discussion (Lines 437-439): “Having controlled for confounding effects of biological and technical variables, we consider that this misclustering is consistent with the nature of decidualization and its inherent variability.”

3. Additional methodological information is needed for long-term reproducibility of analyses.a. For reproducibility of code and analyses, all analytical code for this study should be deposited in a repository such as Github or made available as a Supplementary file.

We also think that the reproducibility of our analysis is a basic scientific focus. This detailed information is specified in the revised manuscript, including that all sequencing data are available for download from the Gene Expression Omnibus (GEO; GSE172381). The software packages and their versions used in the analyses here are provided in the text, and the code needed to use the packages is publicly available on GitHub or Bioconductor (https://bioconductor.org/packages/release/bioc/html/edgeR.html).

See point #1a.

b. Please include all version numbers for all used software (e.g. R, etc.) packages and R packages (e.g. edgeR, goana, etc.), as well as all command parameters where relevant. The same should be applied to annotation databases (i.e. GO used here), and if a version number doesn't exist, date of access should be provided.

Thank you for this observation. We have included this information in the Materials and methods. See point #1a.

c. There is no mention of the softwares (nor versions) used to trim RNA-seq reads, to map to the hg19 genome or to aggregate counts to genes. All used software should be clearly named and their use described with all necessary options for reproducibility.

Thank you for this observation. We have included this information in the Materials and methods. See point #1a.

d. Line 84-88/Figure 1A, were the subtypes of "control" samples (pre vs. full-term) equally distributed in training vs. testing? Indeed, although the authors argue that the subtype made no difference on transcriptional profiles (i.e. Sup Figure 1A PCA), this reviews sees a general (if not clean separation) on PC1, that would probably be enhanced if sources of spurious variation (see point #2) were taken into account. In any case, the data provided cannot be used to support the statement "demonstrated an absence of clustering" (line 80), since the statistical tests used are not meant to prove the null hypothesis but to reject it when possible (i.e. confusion between type I and II error). Please amend the analysis to reflect this.

We agree with the reviewer that this preterm versus full-term analysis should be better explained. Our null hypothesis is that there is** **no differential gene expression across the two subtypes of control samples (preterm versus full-term). We performed a differential gene expression analysis with a proper FDR cutoff for multiple testing comparing preterm versus full-term, revealing that there was no statistical evidence of transcriptomic changes associated with the gestational week at delivery in our set of samples. Thus, gene expression patterns do not provide evidence for rejecting the null hypothesis of “no difference in gene expression between women with preterm or full-term labor who had never had preeclampsia”. This result is shown in the volcano plot in Figure supplement 1A In addition to this plot, we include a PCA because it is a technique for reducing the dimensionality of datasets, increasing interpretability but at the same time minimizing information loss. Thus, we consider it interesting for readers to keep the PCA figure in the manuscript to visualize variance among samples. However, we rewrote this part of the results to clarify that there were no significant differences between control groups. Furthermore, we would like to highlight that the two control subtypes were represented in both subsets of analyzed samples.

We rewrote this part of the Results to clarify this statement (Lines 248-262):

“Controls included women who had a preterm birth with no signs of infection (n=8) and women who gave birth at full term with normal obstetric outcomes (n=8). Transcriptomic profiles were compared by differential expression analysis, revealing no significant changes in the endometrial transcriptome between preterm and term controls [false-discovery rate (FDR) ≥0.05] (Figure 1—figure supplement 1A). Principal component analysis (PCA) supported that there was no underlying pattern of distribution depending on gestational age at delivery (Figure 1—figure supplement 1B). Once we ruled out bias on controls, we randomly split samples into two cohorts, a training set (70%) and a test set (30%) (Figure 1A).”

e. cDNA library cannot be validated for RIN (line 330), as only total RNA samples can be. The entire method section needs to be completely rechecked and rewritten for accuracy to avoid this kind of errors.

According to this recommendation, the Materials and methods section has been reviewed and rewritten when accuracy was required.

The Materials and methods section was updated (Lines 134-136): “cDNA libraries were quantified using an Agilent D1000 ScreenTape in a 4200 TapeStation system (Agilent Technologies Inc, Santa Clara, CA). Libraries were normalized to 10 nM and pooled in equal volumes”. See point #1 for more changes in the Materials and methods section.

f. All used QC filters should be stated explicitly in the methods (see lines 338-342).

We agree with this suggestion, and we have included the information in the manuscript.

Information included in the Materials and methods (lines 148-150): “Quality control filters in each program were used following the software package recommendations, and reads were filtered by mapping quality greater than 90%.”

4. Since RT-qpCR is known to be less sensitive and more prone to normalizing biases (due to the choice of control genes which may vary themselves) than RNA-seq (which performs unbiased normalization at the transcriptome-wide level), it is unclear why validation was performed on the same samples that were processed by RNA-seq (line 375-389). Unless the authors want to include RT-qPCR data on an INDEPENDENT cohort of patients, these results are circular and shouldn't be included in a revised manuscript. If RT-qPCR on a new cohort is included, make sure to include information on which normalizing amplicons were used for δ CT calculation.

We agree with the reviewer about the less sensitive value of RT-qPCR than that of RNA-seq. Following this reviewer’s suggestion, we removed the RT-qPCR data to validate our RNA-seq results from the same cohort of patients.

We removed these data in the revised manuscript. Specifically, data were removed from the Results (Lines 285-287; Lines 298-300). The Materials and methods section was updated (Lines 201-205) “Gene expression of *IHH*, *MMP9*, *MSX2*, *ESR1*, and *PGR* isoforms in the endometrial tissue from a subset of women with prior sPE (n=13) and controls (n=9) was obtained by RT-qPCR. Specific primers for each gene are described in Supplementary file 3.” Figure 2C, Figure 2D, and Figure supplement 2 were removed, and Supplementary file 3 was updated.

5. Can the authors explain why 3 specific samples (C20, 21, 22; line 147-152) do NOT cluster appropriately with their "signature"? Did they have any biological specificities (see need for sample-by-sample information as raised in point #2)? The fact that 3 control samples cluster with the sPE samples somewhat invalidate the signature as a signature of sPE, and may be the result of improper correction for batching or other technical (or irrelevant biological) noise. Please discuss explicitly what may be happening here in a revised manuscript.

We appreciate this comment. The new Supplementary file 2 (containing sample-by-sample information raised in point #2) shows that there was no biological or technical variable that allowed for the identification of the potential source for this misclustering. We suspect that this was due to the complexity of decidualization biology, which is explained in the Discussion section (lines 274–282). Decidualization is a highly dynamic process that shows interindividual variability. In addition, endometrial maturation starts around spiral arteries and extends to the entire endometrium. Consequently, random spatial sampling could affect the proportions of the decidualized cell types during biopsy collection. In fact, other authors, such as Munchel et al., 2020 (PMID: 32611681), have reported a misclustering of controls and PE cases using other approaches. This result is expected due to the high variability inherent to the pregnant human population, PE patients, and the endometrial cycle.

We decided to improve the explanation of this misclustering in the Discussion. Details related to Supplementary file 2 have been included in the Discussion section (Lines 437-439):

“Having controlled for confounding effects of biological and technical variables, we consider that this misclustering is consistent with the nature of decidualization and its inherent variability.”

Reviewer #2 (Recommendations for the authors):In sum, I feel that this paper brings some interesting insights on the decidua status and preeclampsia; the degree of novelty is nevertheless unclear to me, and the clinical application seems far-fetched. There are several points that may be better presented and discussed.Detail of the recommendations to the authors:1. The study of the decidual transcriptome in preeclampsia has also already been performed by other teams in the past, rather at the moment of the disease than later. For instance, the paper of Mari Loset (Am J Obst Gynecol, 2011), is, I feel, an important base for comparison with the results presented here. So much so, that it used the same type of expressional approaches, and detected differential transcripts and cascades. I feel that the authors should analyze their own datasets in light of this type of seminal papers and discuss the commonalities and differences found.

Thank you for your comment. Loset’s work provided interesting insights of genetic canonical pathways and gene–gene interaction networks in decidua basalis from preeclamptic pregnancies at the end of pregnancy, as we previously did [Garrido-Gómez T. et al., 2017 (PMID: 28923940)]. The lack of coincidence between the transcriptomes of decidua during the second and third trimesters of pregnancy and decidualized endometrium in the secretory phase was expected. At the time of the decidua sampling in Loset’s paper, decidualization has already occurred, the decidua has been formed, and PE has been clinically manifested. Therefore, the pattern of gene expression in this tissue may be very different compared to cyclic decidualization without pregnancy or disease. However, following the reviewer’s suggestion, we compared our 859 DEGs with those 455 DEGs from Loset M et al., 2011. We found a modest overlap that included 28 genes, which is consistent with the expected outcome. We also compared our results with the dataset of decidua basalis and decidua parietalis dysregulated genes (79 and 227 DEGs, respectively) in sPE from our group [Garrido-Gómez T. et al., 2017 (PMID: 28923940)]. We found an overlap of 4 DEGs with decidua basalis and 14 DEGs with decidua parietalis. These results are consistent with Rabaglino MB et al., 2019, who performed multiple comparisons, including a comparison of the transcriptome of decidualized endometrial stromal cells from women who experienced sPE (PE-DEC) with the transcriptome of decidua basalis (PE-DB) and parietalis (PE-BP) from the placental bed of preeclamptic patients after delivery (PMID: 31356122). They found that there was little, if any, overlap of differentially expressed genes (DEGs) between PE-DEC and decidua specimens. Moreover, the decidual transcriptome at delivery differs from the decidual tissue of chorionic villi samples collected at ∼11.5 weeks. The authors explain, “These discordances may not be unexpected given the timing of the sample procurement—i.e., PE-DB (decidua basalis obtained from placental bed biopsy) and PE-BP (basal plate decidua obtained from delivered placentas) were procured during clinically active disease that by itself seems likely to perturb decidual gene expression, thus perhaps reflecting consequence rather than cause of disease.” Accordingly, our previous studies demonstrated impaired in vitro decidualization of ESC isolated from delivered placentas, consistent with the concept of endometrial antecedents of sPE (PMID: 28923940).

We agree with the authors that the lack of concordance between transcriptomes may be due to different type of samples (decidua and decidualized endometrium), and different timing of disease (samples at the end of the affected pregnancy versus samples years after) must show large differences in expressed genes. Thus, we did not include this type of comparison in the manuscript. Finally, in the manuscript, we include our own in vitro dataset because we wanted to test the abundance of new transcripts revealed for the first time in our in vivo bulk tissue approach. Also, we include the single-cell results because they were obtained via an in vivo approach using endometrial biopsies from healthy women across the endometrial cycle. Thus, these data could help identify which of the genes that were found to be dysregulated in women who experienced PE are potentially expressed by endometrial stromal cells. Altogether, these comparisons provide new data that we consider interesting due to its novelty.

Discussion (Lines 417-429):

“Previous reports of the decidual transcriptome in PE and sPE (10, 43), revealed the gene expression profile associated with the condition at the time of delivery. Here, we analyszed samples collected years after the affected pregnancy; thus, it could be interesting to find dysregulated genes in common among these approaches. We compared our transcriptomic results and the previously reported deysregulated decidual genes, obtaining 5.1% (10) and 6.2% (43) overlap in affected genes. These discordances may not be unexpected and are consistent with recent results (42). The decidua basalis transcriptome at delivery was compared with the transcriptome of decidua at ⁓11.5 gestational weeks and in vitro decidualized hESC from women who experienced sPE. Both analyses revealed little, if any, overlap between molecular signatures. In contrast, the signature encoding in vitro DD of hESC years after pregnancy overlapped significantly with the dysregulated profile found in decidual samples at the beginning of pregnancy (42). Thus, decidual gene expression patterns during the clinically active disease largely differ from those observed during endometrial decidualization at the end of the menstrual cycle and early pregnancy, perhaps reflecting a consequence rather than the origin of sPE.”

2. In the overrepresented pathways in the Loset's work, the regulation by ESR1 and PGR was not obvious. Does it mean that analyzing samples 3-4 years after the event (normal or pathological pregnancy) leads to remnants of the disease at the uterine level, or is there a genetic predisposition? This is a question that cannot be truly addressed by the present dataset, and is important if the objective is to define markers, as stated below.The introduction gives the necessary details to understand the question raised, finding a signature that could help assessing the risk of preeclampsia.

Thank you for highlighting this interesting issue for discussion. In Loset’s work, the authors were focused on canonical pathways that could be affected by dysregulated genes in PE, which was valuable to increase the knowledge of PE pathogenesis. However, important differences between Loset’s work and our work need to be mentioned to answer this point. In Loset’s work, the authors did not examine in-depth the origin of the identified gene dysregulation. Biological processes or pathways tend to be quite specific, making it difficult to find obvious associations. We went one step further trying to investigate the association with hormone receptor activation due to the key role of ovarian hormones in decidualization. We tested this hypothesis, and this association became obvious from our results. However, Loset et al., did not address the same question.

Regarding the question “Does it mean that analyzing samples 3–4 years after the event (normal or pathological pregnancy) leads to remnants of the disease at the uterine level or is there a genetic predisposition?”, we agree with the reviewer that it could not be answered. Other analyses are needed, such as genetic variant analysis, to know if there is a genetic predisposition. We acknowledge the existence of this altered decidual transcriptome in sPE years after delivery, but there are other works that demonstrate that a decidualization defect occurs at both the end and the beginning of pregnancy (PMID: 28923940, PMID: 31356122). In addition, endometrial decidualization as a primary factor in pregnancy health and evidence to support the role of deficient decidualization in the origin of PE were presented last year (PMID: 32521725, PMID: 33007270). Therefore, genetic predisposition for a decidualization defect is plausible on this basis.

Discussion (Lines 498-501):

“However, the primary driver of the predisposition to undergo decidualization resistance and its link with the main risk factors of sPE remain to be determined. Our work is an important step toward the development of new strategies that enable early assessment of risk for sPE and might prompt new therapeutic strategies to treat this enigmatic pathological condition.”

3. However, the genetic part of preeclampsia is estimated by a heritability of ~55%. The genetic decomposition of this heritability presented by Cnattingius and coworkers (2004) indicate that only part of this genetics is connected to the maternal genetic background (which concerns two organs: the uterus and thus the decidua) and the placenta for half of its genetics. Part of the genetics of preeclampsia is associated only to the paternal genetics (estimated at ~1/3 of the heritability), therefore, it is clear that the risk cannot systematically be found by analysis of the female (uterine-decidua) expression profile only. Alternatively, the difference observed by the authors may be a consequence of the previous preeclampsia not based on a predisposition, but if this is the case, it limits considerably the usefulness of the markers found since preeclampsia, and in particular severe preeclampsia is at 75% a disease of the first pregnancy. The authors should recognize this as an important limit as an early prenatal screening strategy.

Preeclampsia is a syndrome in which different condition subtypes might coexist. Thus, to advance in its understanding, we have focused on the maternal contribution to severe preeclampsia through defective decidualization. Also, we are not claiming that our findings are related to heritability (DNA), but rather to the pathogenesis of the decidualization defect (RNA expression) involved in the origin of severe PE (PMID: 32521725; PMID: 33007270).

Discussion (lines 493-496):

“Preeclampsia is a syndrome in which different condition subtypes might coexist; here, we focused on the maternal contribution to sPE through DD. We are not claiming that our findings are related to sPE heritability (DNA), but instead, our findings are related to the pathogenesis of the decidualization defect (RNA expression) involved in the origin of PE (12, 22)”.

The initial analysis is based upon the endometrial tissue of 24 women that had a severe preeclampsia, 16 controls (8 preterm and 8 term births). Gestational age at delivery had apparently no influence on the 'control' groups. Since the samples were not collected at the end of gestation, which means that the occurrence of preterm may be connected to the placenta and not to the maternal uterine situation.4. Nevertheless, there are only 8+8 samples so it is not possible to prove that in some case a decidual expression is not involved. In addition, using PCA to check that there is no clustering is not enough to certify that there are no differentially expressed genes that separate the two groups, it means only that the number of differential genes compared to the mass of genes that are not differentially expressed is not enough to cluster the groups, which may be the case if the percentage of genes changed is small. There are some published evidences that recurrent spontaneous abortion, pre-term birth and preeclampsia share common mechanisms. The observation here seems to indicate that the uterus of women with PTB is identical to the one of term pregnancies. From the clinical data, the authors acted wisely in taking PTB without hypertension. But it is not clear from the data that the women that had PTB had a recurrent occurrence in this (not mentioned in Table S1).

We thank the reviewer for the opportunity to improve the clarity of this point in our manuscript. We performed a gene expression analysis comparing preterm with full-term labor, revealing that there were no transcriptomic changes associated with the gestational age at delivery in our set of samples. This result is shown in the volcano plot in Figure supplement1A. In addition to this plot, we include a PCA because it is a widely used conventional method for reducing data complexity and visualizing the variance among samples. Thus, gene expression patterns do not provide evidence for rejecting the null hypothesis of “no difference in gene expression between women with preterm or full-term labor who had never had preeclampsia”. We agree with the reviewer that the sample size is limited to state conclusions such as “the uterus of women with PTB is identical to the one of term pregnancies”, but this is not our objective nor our intention. Our hypothesis is to demonstrate that patients that experience previous sPE have transcriptomic changes in the endometrium during the late secretory phase. To find the altered transcripts, we compared a case group with a control group, which was heterogeneous for the gestational age at delivery; but, testing differences due to preterm or full-term labor are negligible in controls compared to the differences induced in the cases that experience sPE.

We have rewritten this part of the Results to clarify this statement (Lines 248-256):

“Controls included women who had a preterm birth with no signs of infection (n=8) and women who gave birth at full term with normal obstetric outcomes (n=8). Transcriptomic profiles were compared by differential expression analysis, revealing no significant changes in the endometrial transcriptome between preterm and term controls [false-discovery rate (FDR) ≥0.05] (Figure 1—figure supplement 1A). Principal component analysis (PCA) supported that there was no underlying pattern of distribution depending on gestational age at delivery (Figure 1—figure supplement 1B).”

5. In addition, the absence of DEG genes visible (Supplementary Figure 1B) is presented against FDR and not p-value which is unusual and I think a bot too stringent. Even random samples should give some significant genes (one out of 20) Using FDR leads to no significant genes, which is not a complete surprise. However, it does not mean that individual genes are not relevant for the difference between the two situations analyzed. Another interesting approach should be to check for enrichment of pathways using GSEA approaches, that do not rest on the establishment of thresholds that are always arbitrary. In sum excluding totally the existence of DEG between the two groups seem a bit rapid.

The p-value adjustment method for multiple comparisons was an FDR with a cutoff of 0.05. Thus, we used the p-value adjusted for this analysis, and we applied the same method for the gene expression analysis between controls and sPE. We are comparing more than 18,000 genes between groups. With any p-value adjustment, we will obtain significant differences, but many of them would be false positives just by pure random chance. In this regard, FDR retains more significant p-values while increasing non-significant p*-*values than the Bonferroni method. Thus, we selected a less conservative adjustment method to identify differences if they exist, maintaining the FDR at 5%. In fact, the most preferable approach is controlling FDR as it not only reduces false positives but also minimizes false negatives (more details can be found in PMID: 30124010).

6. In the legend of Suppl figure S1, I do not understand 'Plot based on 728 genes labeled as fc (blue) and 17,748 genes labeled as none (purple)'. Nevertheless, it is admissible that the differences due to preterm or term are negligible compared to the differences induced by sPE.The decomposition of the samples between sets for simulations is unusual in the field, but mathematically sound. The authors discovered 859 DEG at a threshold of 2-fold, and some genes (9) were validated by qRT-PCR.

Labels show the two criteria that we used to define the differentially expressed genes: adjusted p-value (FDR<0.05) and fold change (FC≥2). Blue dots represent genes that were not statistically different in expression, but that demonstrated an expression change (FDR≥0.05 and FC ≥2). Purple dots show those genes that were not significantly different in expression, with only a slight shift in expression (FDR≥0.05 and FC<2)

We rewrote the legend of Figure supplement 1 (752-758): “Figure 1—figure supplement 1. Transcriptomic analysis of control samples based on gestational age at delivery. (A) Volcano plot showing that there were no significant DEGs between controls according to gestational age at delivery. Labels show the two criteria that we used to define the differentially expressed genes: Adjusted p-value (FDR<0.05) and fold change (FC≥2). The plot is based on 728 genes labeled as “Not significant-FC≥2” (blue) and 17,748 genes labeled as “ Not significant-FC<2” (purple). (B) A PCA based on 18,476 genes, after removing genes with low expression, does not demonstrate clustering based on gestational age.”

7. It would be nice to analyze the data not only relative to a threshold of induction, but taking the complete dataset and using GSEA-like approaches to see whether they are consistent with the gene clusters found by threshold-volcano plots analyses.

We did not use an analysis of the data only relative to a threshold of induction, we took the complete dataset and used the edgeR package to obtain those genes that were significantly differentially expressed between groups (FDR<0.05). Once those DEGs were obtained, we emphasized the high fold change between groups (FC≥2 and FC≥4).

For the first filtering step, we used the exactTest function (edgeR package), which is recommended to make comparisons on datasets with a single-factor design, allowing us to achieve a global exploratory approach. This function uses the p-value adjustment method FDR (“False Discovery Rate”), and a cutoff of 0.05 was applied to identify DEGs. Once we obtained the significantly differentially expressed genes, we applied a second filtering step, fold change, to select those genes where differences were large enough to be biologically meaningful. To show visually the proportion of genes that pass through the filters (after analyzing the complete dataset), we used a volcano plot. We specified these two filters in each line, and we refer to DEGs and in the section “Differential expression analysis” from the Material and Methods, but we improve the information provided about the statistical analysis.

– Specifically, we rewrote “Quality control and pre-processing data” and “differential expression analysis” sections of the Materials and methods. These sections were replaced by a new section titled “RNA-seq analysis” (Lines144-167):

“Reads were mapped to the hg19 human genome transcriptome using the STAR (version 2.4.2a) read aligner (25). FastQC (version 0.11.2) was used to determine the quality of FASTQ files. The manipulation of SAM and BAM files was done with the software SAMtools (version 1.1) (26). To count the number of reads that could be assigned to each gene, we used HTSeq (version 0.6.1p1) (27) and BEDtools software (version 2.17.0) (28) to obtain gene coverage and work with bedFiles. Quality control filters in each program were used following the software package recommendations, and reads were filtered by mapping quality greater than 90%.. Transcriptomic data were deposited in the Gene Expression Omnibus database (accession number GSE172381). The Bioconductor package edgeR (version 3.24.3) (29) was used to analyze differentially expressed genes. The trimmed mean of M-values normalization method was applied to our gene expression values. The exactTest function was used to find differentially expressed genes between groups. The p-value adjustment method was FDR with a cut-off of 0.05 (FDR<0.05). Once p-value was adjusted, significant deregulated genes with log2-fold-change ≥1 (FC≥2) were selected to perform gene ontology analysis and to formulate the signature encoding DD. edgeR analysis was carried out in R version 3.5.1. A volcano plot was created to visualize DEGs. For a better overview, we distinguished significant (FDR<0.05) and not significant (FDR≥0.05) DEGs with a high (FC≥2) or low fold-change (threshold FC<2).”

– We improved the explanation about the statistical analysis applied in the Results section (Line 270-273):

“Transcriptional analysis in the training set was performed by comparing gene expression patterns in sPE (n=17) and controls (n=12). This comparison revealed 859 differentially expressed genes (DEGs) based on FDR<0.05 and at least two-fold differential expression between groups (FC≥2)” and (Line 293-295): “Eighteen genes were similarly differentially expressed with at least a two-fold change between groups (FDR<0.05 and FC ≥2) (Figure 2A).”

– The legend of Figure 1B was also updated (lines 672-674):

“not significant-lowFC (FDR≥ 0.05, FC<2); not significant-highFC (FDR≥0.05; FC≥2); significant-lowFC (FDR<0.05; FC< 2); significant-highFC (FDR<0.05; FC≥2).”

– Supplementary file 3 changed its name by Figure 1-Source data 1 and p-value, FDR values (corrected p-value), log FC, and FC, and we included how the criteria were applied in the legend (779-782):

“Figure 1-Source data 1. The 859 statistically differentially expressed genes (FDR<0.05) with at least two-fold change (FC≥2) in sPE versus control cases obtained from RNA-seq analysis.”

– The Supplementary file 4 changed its name by Figure 3- Source data 1 and legend was updated (lines 783-784):

“Figure 3- Source data 1. List of genes selected as the defective decidualization signature in sPE (166 DEGs with an FDR of <0.05 and an FC of ≥4).”

Previous studies by the authors evaluate gene expression differences in decidualization either from human endometrial stromal cells (hESCs) from women with sPE versus normal. The authors identified 18 genes differentially expressed in vivo and in vitro in sPE compared to control.8. The authors do question the observation of having only 18 common genes between in vivo and in vitro, out of 129 or 859 (in vitro – previous study- and in vivo -present study- respectively), and base them upon cell composition that is much more complex in vivo, which is reasonable. However, in the search for a signature, which is one of the justifications of the work, it could mean that the in vivo dataset could contain more 'relevant' genes compared to a in vitro model. The comparison with the results of Wang, which is much better (and focused better on a simple cell model) on the single-cell transcriptome may belong rather to a discussion rather than a results part of the paper. The validation by qRT-PCR is good, but the choice of the genes leads to a very high correlation (2D), that may be due to the use of only one induced gene (and only 5 genes used). About the actin as a control housekeeping gene, it is not sure that it is the best reporter gene in the uterine context. Generally, it is advised to normalize against the geometric mean of two to four different reporter genes.A selection of 166 highly deregulated genes (>4 fold) is then selected, and were shown to be enough to separate efficiently the samples, which is not surprising, given that this corresponds to a semi-supervised analysis from genes found differential between the two gene sets.

The comparison with the results of Wang (PMID: 32929266) was included in the Results section because this dataset was obtained from an in vivo approach using endometrial biopsies from healthy women across the endometrial cycle and identified gene expression patterns at a single-cell level. The analysis used in the current manuscript is from bulk tissue. Therefore, combining our dataset with the dataset from Wang provides insight into which of the genes that were identified as dysregulated in women who experienced preeclampsia may be expressed by endometrial stromal cells specifically. The data from this comparison analysis increases the specificity of our results.

We decided to remove the qRT-PCR data in the revised manuscript following the recommendation of Reviewer #1. The Reviewer suggested that RT-qPCR is less sensitive than RNA-seq and reported circular results because it was applied to the same cohort of patients.

The 166 highly deregulated genes were obtained comparing gene expression patterns between cases and controls from the training set. Then, these genes were applied to the test set of samples. This revealed an efficient separation of groups, validating the results observed in the training set.

The Materials and methods section was updated (Lines 201-205):

“Gene expression of *IHH*, *MMP9*, *MSX2*, *ESR1*, and *PGR* isoforms in the endometrial tissue from a subset of women with prior sPE (n=13) and controls (n=9) was obtained by RT-qPCR. Specific primers for each gene are described in Supplementary file 3.”

We removed these data in the revised manuscript. Specifically, data were removed from the Results (Lines 285-287; Lines 298-300). Figure 2C, Figure 2D, and Figure supplement 2 were removed, and Supplementary file 3 was updated.

9. The identification of Estrogen/progesterone receptor is not a surprise, when uterus function is concerned. I would suggest that the authors complete their analyses using network analyses such as provided by the combination of Stringdb and Cytoscape. This would help to visualize the pivotal position of ESR1 and PGR more clearly, or maybe to find other important hubs that were overlooked in their current study. Also using GSEA or other tools on transcription factor binding sites databases, showing the actual involvement (through measuring the enrichment, and calculation of FDRs) of the Estrogen Responsive Element and Progesterone Responsive Element would be an essential element that must be shown to prove a genuine enrichment in these cascades

Following the reviewer’s recommendation, the hub genes were determined by overlapping the top 20 genes obtained using two topological analysis methods of the cytoHubba plugin, maximal clique centrality (MCC) and maximum neighborhood component (MNC). In addition, genes responsive to estrogen and progesterone have been identified using the database of Human Transcription Factor Targets. Details are included in the revised manuscript.

– Materials and methods (Lines 195-198):

“Hub genes were extracted using the maximal clique centrality (MCC) and maximum neighborhood component (MNC) of the cytoHubba plugin (32). The overlapping genes identifiedy by the two topological analysis methods were selected as the hub genes.”

– Results (Lines 346-348):

“Regarding target genes of ER1 and PR, the database of Human Transcription Factor Targets (hTFtarget) reported 13 genes responsive to ER1 and 31 target genes modulated by PR, based on epigenomic, ChIP-seq, or motif evidence (36)”. Lines 358-361: “The hub genes were determined by overlapping the top 20 genes obtained using two topological analysis methods in the cytoHubba plugin (31), maximal clique centrality (MCC) and maximum neighborhood component (MNC). Sixteen genes were selected, all of which were downregulated.”

Reviewer #3 (Recommendations for the authors):There is much debate on whether the risk of PE is higher/lower in women with male vs. female fetuses. Additionally, the maternal response to pregnancy differs based on the sex of the fetus. The manuscript should therefore report on this variable and whether it impacts the results of the analysis.

We included 24 women who experienced sPE, of whom 9 had a female fetus and 11 had a male fetus (data for two patients were unavailable). The PCA and heat map in (Author response image 2) represent this variable, showing a heterogeneous distribution of samples [female (yellow) and male (green) fetuses]. Thus, we do not include this result in our manuscript since we do not focus on sPE risk factors, and we the sex of the fetus in our limited series does not impact the results of our transcriptomic data.

**Author response image 2. sa2fig2:** Validation of the DD fingerprint in sPE. (A) PCA based on 166 genes included in the fingerprinting in the training set. Each sample is represented as a colored point (blue, control; orange, sPE). (B) Heatmap dendogram of expression of the 166 genes included in the final fingerprinting for each sample of the training set (control, n=12; sPE, n=17). Sex of the fetus is represented by color (yellow, female; green, male). (**C**) PCA based on the fingerprinting in the test set. Each sample is represented as a colored point (blue, control; orange, sPE). (D) Heatmap dendogram of expression of the 166 genes included in the final fingerprinting for each sample of the test set (control, n=4; sPE, n=7). Sex of the fetus is represented by color (yellow, female; green, male).

Please clarify whether biopsies were done for clinical or research purposes only.

Biopsies were performed for research purposes only. This study was approved by the Clinical Research Ethics Committee of Hospital La Fe (Valencia, Spain) (2011/0383).

We clarified this observation in the manuscript (Line 92-94):

“Endometrial samples were obtained for research purposes during late secretory phase in 24 women who had developed sPE in a previous pregnancy and in 16 women with no history of sPE with full term (n=8) and preterm pregnancies (n=8) as controls.”

Why did the authors choose to use hg19 as their reference genome and not the better annotated hg38?

The latest build of the human reference genome, commonly nicknamed hg38, greatly expanded the repertoire of ALT contigs. These represent alternate haplotypes and have a significant impact on our power to detect and analyze genomic variation that is specific to populations that carry alternate haplotypes. Thus, this version is strongly recommended for analyzing genetic variation. However, we were interested in analyzing gene expression patterns instead of sequence variation. The existence of different versions causes confusion, and part of the problem is that many bioinformatic tools fail to enforce consistent use of a specific reference. This allows the unwary user to switch reference genomes halfway through a project without realizing that their comparisons suddenly become worthless (because, for example, now all the positions are shifted by some coordinate index). Hg19 is still the most widely used version; consequently, many studies, and even clinical databases and bioinformatic tools, are based on it. Thus, version hg19 is worthwhile for gene expression analysis and facilitates the use of our dataset by the scientific community.

We included these details in the Materials and methods (lines 144-145):

“Reads were mapped to the hg19 human genome transcriptome using the STAR (version 2.4.2a) read aligner (25).”

Methods (line 321) – What RIN value was defined as appropriate for making RNAseq libraries.

RIN values ranged from 4.9 to 9.2. This detail has been included (Line 124).

I cannot find anywhere where DD is defined. Can the authors please clearly define what they mean when they use DD?

Thank you for this observation to clarify this abbreviation in the manuscript. We defined the term in Lines 63-65. The abbreviation is now included:

“Defective decidualization (DD) entails the inability of the endometrial compartment to undertake tissue differentiation, leading to aberrations in placentation and compromising pregnancy health (12)”.

Introduction – PE also significantly contributes to maternal mortality, not just infant.

We have included the PE contribution to maternal mortality in this sentence (line 34).

Change in line 49-51:

“PE is characterized by the onset of hypertension, proteinuria, and other signs of maternal vascular damage that contribute to maternal and neonatal mortality and morbidity (1)”.

Figure 3B – what is the axis label?

The axis label refers to “Enrichment index”, which is calculated by −log(p-value). In Figure 3B, the axis label has been included in the figure and is explained in the legend.

Figure 4 – I believe the authors mean sPE (not Spe).

Yes, we mean sPE. Thank you for the observation. Spe was replaced by sPE in Line 721

Data availability: authors state that the RNAseq data is available for download in GEO.

Transcriptomic data were deposited in the Gene Expression Omnibus database (accession number GSE172381), as specified in the Materials and methods (RNA-seq analysis). However, after this recommendation, we decided to include the accession number in the Results section.

GSE code was included in the Results section (Lines 238-240):

“To identify transcriptomic alterations during decidualization in sPE, we applied global RNA sequencing (RNA-seq) to endometrial biopsies obtained in the late secretory phase from women who developed sPE in a previous pregnancy (n=24) and from controls who never had sPE (n=16) (GSE172381).”

[Editors’ note: the authors resubmitted a revised version of the paper for consideration. What follows is the authors’ response to the first round of review.]Essential Revisions (for the authors):The authors failed to to properly respond to key reviewer concerns, which were considered to be necessary changes by the consensus of reviewer's discussion. Thus, the manuscript cannot be further considered until these points raised by the reviewers are appropriately addressed.

We would like to thank the reviewers and editors for their time and effort devoted to improving our manuscript. We addressed the key reviewer concerns in the revised manuscript, please find bellow our point-by-point answer.

1. The editor and reviewers understood the method used for fold change thresholding. No additional explanation was required. The method is still wrong, as post-FDR fold-change thresholding is the problem that was raised and cannot be used in a rigorous analysis. Thus, the authors should implement one of the previously given options so that this major problem is corrected: (i) get rid of fold-change thresholding completely, or (ii) use a method that does it, while appropriately controlling for FDR (e.g. TREAT).

Following the reviewer suggestion we used the method TREAT to test formally the hypothesis (with associated p-values) that the differential expression is greater than a given threshold. Changes have been implemented along the manuscript since the analysis was rerun with a different method (TREAT instead of ExacTest). Thus, we updated Materials and Methods, results and figures.

2. Similarly, using multiple corrections for GO and KEGG analysis is not a suggestion but a necessity for rigorous analysis, especially when performing so many tests. If term redundancy is a major concern for the authors in the GO database, they are advised to look at the GOSlim framework which was put together to specifically respond to this concern. In any case, enrichment results without FDR control are non-reproducible and not statistically supported.

According to this recommendation, multiple corrections for GO analysis was applied. We rewrote the part of GO analysis in Materials and Methods (lines 170-176) and results (lines 277-292).

3. The authors have not responded to the requirement to upload all code/scripts generated for the study. They refer to the software repository (i.e. edgeR), but not the actual code for analysis. The specific scripts that were written to run all analyses need to be made available as well on GitHub or as a supplement, as per eLife policy.

We made available the scripts that were written to run all analyses. They could be found in Github (https://github.com/mclemente-igenomix/garrido_et_al_2021). We included the sentence “Custom scripts are available on GitHub at https://github.com/mclemente-igenomix/garrido_et_al_2021” in lines 159, 166, and 176. Also, the link was included in the key resources table.